# Lower Bounds for Chain-of-Thought Reasoning in Hard-Attention Transformers

**Alireza Amiri** [1]  **Xinting Huang** [2]  **Mark Rofin** [2]  **Michael Hahn** [2]

## Abstract

Chain-of-thought reasoning and scratchpads have emerged as critical tools for enhancing the computational capabilities of transformers. While theoretical results show that polynomial-length scratchpads can extend transformers' expressivity from $TC^0$ to $PTIME$, their required length remains poorly understood. Empirical evidence even suggests that transformers need scratchpads even for many problems in $TC^0$, such as PARITY or MULTIPLICATION, challenging optimistic bounds derived from circuit complexity. In this work, we initiate the study of systematic lower bounds for the number of CoT steps across different algorithmic problems, in the hard-attention regime. We study a variety of algorithmic problems, and provide bounds that are tight up to logarithmic factors. Overall, these results contribute to emerging understanding of the power and limitations of chain-of-thought reasoning[1].

## 1. Introduction

Chain-of-Thought reasoning (CoT) has become a standard practice for solving hard problems with LLMs, enhancing the capabilities of Transformers (Nye et al., 2021; Wei et al., 2022b) and powering a new generation of state-of-the-art models, such as OpenAI o1 (Jaech et al., 2024) and DeepSeek-R1 (Guo et al., 2025). Models trained under this paradigm are optimized to generate a long CoT before answering any user's request, which significantly elevates their reasoning abilities. However, the use of CoT may substantially increase the number of tokens produced by the model and raise the inference costs (Han et al., 2024), in some cases reaching millions of tokens for a single task (Chollet, 2025). Hence, shortening the generated CoT se-

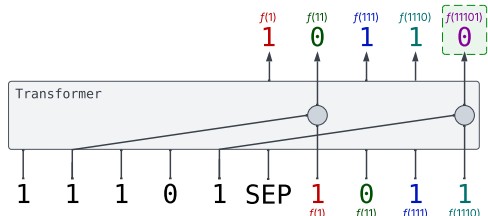

Figure 1: High-sensitivity problems such as PARITY are difficult to learnably and generalizably represent for transformers. Such problems can be solved by CoTs decomposing them into a sequence of local steps. We study the length that such CoTs need to have, as a function of the input length.

quences without compromising quality became an important research direction (Deng et al., 2024). At the same time, there is so far little understanding of the minimal sufficient length of the CoT in a given case; thus the theoretical limits of CoT compression are unclear.

To gain understanding of those limits, we take up the problem of provably bounding the length of CoT sequences sufficient to solve various algorithmic problems with transformers. By providing the lower bounds on the length of the CoT, our paper complements an established line of research on the expressive power of transformers that has focused on the model size when the answer is provided without CoT (e.g. Strobl et al., 2023; Bhattamishra et al., 2020a; Sanford et al., 2024b; Chen et al., 2024).

A simple example is the PARITY function – deciding whether the number of 1's in a bit string is even or odd. Transformers empirically struggle to learn it on longer inputs (e.g. Bhattamishra et al., 2020a; Delétang et al., 2023; Butoi et al., 2024; Hahn & Rofin, 2024), and this difficulty is resolved by a CoT consisting of the parities of increasing prefixes, as in Figure 1 (e.g. Anil et al., 2022). Here, every decoding step depends only on a bounded number of tokens. However, for an input of length $N$, this CoT requires $\Theta(N)$ extra decoding steps, a substantial computational burden.

An interesting and pertinent question is thus whether such a linear-length CoT is optimal, or whether a shorter CoT might exist. The goal of this paper is to *develop explicit and unconditional bounds on the length of CoTs*. This

---

[1]Sharif University of Technology. Work done in part while interning at Saarland University. [2]Saarland University. Correspondence to: Michael Hahn <mhahn@lst.uni-saarland.de>.

*Proceedings of the $42^{nd}$ International Conference on Machine Learning*, Vancouver, Canada. PMLR 267, 2025. Copyright 2025 by the author(s).

[1]Code link: https://github.com/lacoco-lab/scratchpad_bounds

question is analogous to the classical question of bounding the time complexity of Turing machines. We argue that it is of foundational interest in the context of LLM-based systems relying on CoT reasoning (Jaech et al., 2024; Guo et al., 2025).

We study this question in the *unique-hard attention* (UHAT) regime, a popular theoretical abstraction of self-attention, in which every attention head attends to the unique position where attention weights are maximized (e.g. Hahn, 2020; Hao et al., 2022; Barceló et al., 2024; Svete & Cotterell, 2024; Bergsträßer et al., 2024; Yang et al., 2024; Barceló et al., 2025). Importantly, bounds in this regime entail bounds on *realistic softmax transformers operating at fixed precision*, as their expressiveness is upper-bounded by UHAT (Jerad et al., 2025).

In this model, we rigorously prove that the CoT for PARITY described above is optimal up to constant factors (Theorem 4.2). Besides PARITY, we consider three other tasks (MULTIPLICATION, MEDIAN, REACHABILITY) that are also empirically challenging for transformers to solve without CoT. Across tasks, we show that CoT lengths need to scale at least linearly in the size of the original problem. We further show that our bounds are *optimal up to logarithmic factors*. Overall, these lower bounds place broad constraints on the inference-time compute needed to enable transformers to solve these problems.

## 2. Background

### 2.1. Sensitive Functions Require CoTs

Prior work has often studied the abilities of transformers through the lens of *circuit complexity* (e.g. Hao et al., 2022; Merrill & Sabharwal, 2023a; Strobl, 2023; Feng et al., 2023; Li et al., 2024; Chiang, 2025; Merrill & Sabharwal, 2023b). Transformers express a subset of the circuit complexity class $TC^0$, which covers problems solvable by bounded-depth threshold circuits (Merrill & Sabharwal, 2023b; Strobl, 2023; Chiang, 2025). By the unproven conjecture $TC^0 \neq NC^1$, many problems, such as graph reachability, are thus not solvable by transformers without CoTs. Hence, the success of CoT has been linked to its ability to expand expressiveness beyond $TC^0$ (Feng et al., 2023; Merrill & Sabharwal, 2024a; Li et al., 2024).

However, CoT is empirically beneficial even for tasks well in $TC^0$, such as PARITY (Anil et al., 2022): PARITY is expressible in $TC^0$ and also by certain models of transformers (Chiang & Cholak, 2022; Kozachinskiy, 2024), but practical transformers consistently struggle to learn it via SGD in long inputs (e.g. Bhattamishra et al., 2020a) unless a CoT is provided (e.g. Anil et al., 2022). Hahn & Rofin (2024) show that this fact can be explained in terms of the loss landscape: Transformers require very sharp minima to represent

functions that, like PARITY, are simultaneously sensitive to every input bit, and hence do not practically learn them on long inputs; sharpness is starkly reduced in the presence of a CoT. In fact, the result of Hahn & Rofin (2024) extends beyond PARITY, and is grounded in a foundational concept in the analysis of Boolean Functions, *sensitivity*. For a boolean function $f : \{0,1\}^* \rightarrow \{0,1\}$, the *sensitivity* counts how many Hamming neighbors have the opposite output:

$$s(f,x) = \left| \{i = 1, \ldots, |x| \text{ such that } f(x) \neq f(x^{\oplus i})\} \right| \tag{1}$$

where $x^{\oplus i}$ is obtained from $x$ by flipping the $i$-th bit. The *average sensitivity* is defined as the average over the Hamming cube, at string length $N$ (e.g. O'Donnell, 2014):

$$as_N(f) = \frac{1}{2^N} \sum_{x \in \{0,1\}^N} s(f,x) \tag{2}$$

In PARITY, flipping any bit flips the output, hence $as_N(\text{PARITY}) = N$. Hahn & Rofin (2024) show that such linear growth of sensitivity with input length produces high sharpness in transformers' loss landscapes. This unifies a string of empirical results finding that transformers have an inductive bias towards low average sensitivity (Bhattamishra et al., 2023; Vasudeva et al., 2024; Abbe et al., 2023). Hahn & Rofin (2024) applied this result to the PARITY function; but, in this paper, we identify a range of other tasks also facing linear growth of sensitivity, including tasks within $TC^0$ (MULTIPLICATION and MEDIAN), and a task conjectured not to be in $TC^0$ (REACHABILITY). We thus overall

*focus on algorithmic tasks with $as_N(f) = \Theta(N)$.*

including both tasks within $TC^0$, and tasks conjectured to be outside it.

### 2.2. Model of Transformers

The theoretical literature on transformers has developed various formal abstractions of transformers. In this paper, we study the regime of *unique hard attention* (UHAT), a popular theoretical model of self-attention, where every attention head attends to the unique position where attention scores are maximized (e.g. Hahn, 2020; Hao et al., 2022; Barceló et al., 2024; Svete & Cotterell, 2024; Bergsträßer et al., 2024; Yang et al., 2024; Barceló et al., 2025). UHAT is an appealing modeling choice, both because strong techniques for proving lower bounds are available (see Section 3), and because interpretability work shows that language models heavily rely on heads focusing their attention on few positions (e.g. Cabannes et al., 2024; Olsson et al., 2022; Clark et al., 2019; Voita et al., 2019; Ebrahimi et al., 2020).

We now introduce the relevant notions and notation. We assume a finite alphabet $\Sigma$, with token embeddings $e(\sigma) \in \mathbb{R}^d$. There further are positional encodings

$\boldsymbol{p}_1, \boldsymbol{p}_2, \boldsymbol{p}_3, \ldots, \boldsymbol{p}_{n_{max}} \in \mathbb{R}^d$, where $n_{max}$ is the maximal context size of the transformer. We consider an input string $x \in \Sigma^*$, with length $|x| = N$. We define the activations $\boldsymbol{y}_i^{(k)} \in \mathbb{R}^d$ at position $i$ of the $k$-th layer ($k = 1, \ldots, L$) as follows. The zero-th layer consists of token and positional encodings: $\boldsymbol{y}_i^{(0)} := e(x_i) + \boldsymbol{p}_i$ ($i = 1, \ldots, N$). In each layer $l = 1, \ldots, L$, we first compute attention scores for the $h$-th head ($h = 1, \ldots, H$):

$$a_{i,j}^{(k,h)} = (\boldsymbol{y}_j^{(k-1)})^T K_{k,h}^T Q_{k,h} \boldsymbol{y}_i^{(k-1)}$$

where $K_{k,h}$ ("key"), $Q_{k,h}$ ("query") are $\in \mathbb{R}^{d \times d}$. In softmax attention, the attention weights $\widehat{a}_{i,j}^{(k,h)}$ are then obtained via the softmax transform. In the UHAT model, these are idealized as one-hot weights:

$$\widehat{a}_{i,j}^{(k,h)} = \frac{\exp(a_{i,j}^{(k,h)})}{\sum_{s=1}^{i} \exp(a_{i,s}^{(k,h)})} \qquad \text{[Softmax]} \quad (3)$$

$$\widehat{a}_{i,j}^{(k,h)} = \begin{cases} 1 & j = \arg_{s \leq i} \max a_{i,s}^{(k,h)} \\ 0 & else \end{cases} \qquad \text{[UHAT]} \quad (4)$$

The UHAT model can be viewed as the limit of the Softmax model when one attention score $a_{i,s}$ far exceeds the others. If more than one $s$ attains the maximal attention score, ties are broken according to some fixed rule (e.g., choosing the left- or right-most match, Yang et al. (2024); Jerad et al. (2025)). The output of the attention block is computed by weighting according to attention weights $\hat{a}_{i,j}^{(k,h)}$ ($j \leq i$)[2], and applying a linear transformation $V$ ("value"); these are then aggregated across heads and combined with a skip-connection:

$$\boldsymbol{y}_i^{(k)} := f^{MLP} \left( \boldsymbol{y}_i^{(k-1)} + \sum_{h=1}^{H} \sum_{j=1}^{i} \hat{a}_{i,j}^{(k,h)} V_{k,h} \boldsymbol{y}_j^{(k-1)} \right)$$
(5)

where $f^{MLP} : \mathbb{R}^d \to \mathbb{R}^d$. For our purposes here, $f^{MLP}$ may be arbitrary.[3] Finally, next-token predictions are made by $T := \boldsymbol{O} \cdot \boldsymbol{y}_n^{(L)}$ for some parameter $\boldsymbol{O} \in \mathbb{R}^{|\Sigma| \times d}$, where we are assuming some numbering of the alphabet $\Sigma$.

### 2.3. Formalizing CoTs

**Definition 2.1.** Let $\Sigma$ be a finite alphabet. Given a function $f : \Sigma^* \to \Sigma^+$ and an alphabet $\Xi \supset \Sigma$, a *chain-of-thought* (CoT) is a map $g : \Sigma^* \to \Xi^+$ ending in the suffix $f(x)$.

---

[2]We assume causal masking in line with standard language model architectures, in order to allow autoregressive generation of the CoT. Our techniques are also applicable to the setting where the input itself is processed with bidirectional attention as assumed by Abbé et al. (2024) (Remark B.7).

[3]Transformers additionally implement layer norm (Ba et al., 2016). Our bounds are robust to such position-wise operations. For instance, layer norm following the MLP can be absorbed into $f^{MLP}$ for the purposes of our theorems.

We write $\Xi_N$ to be the (finite) set of symbols appearing in at least one string $xg(x)$ for $x$ with $|x| \leq N$.

We note that $\Xi$ is not restricted to be finite. Some theoretical work has allowed the CoT vocabulary to grow with the input length (Bhattamishra et al., 2020b; Abbé et al., 2024); our lower-bounds are robust to this.

We now formalize what it means for a CoT to be expressible in UHAT. For maximal generality, we assume a relaxed definition: rather than requiring that a single transformer perform the task across all input lengths (which makes it hard to distinguish unboundedly many positions), we only ask that a *family* of transformers perform the task with a bounded number of layers and heads:

**Definition 2.2.** We say that a transformer $T$ computes the CoT $g(x)$ on input $x \in \Sigma^*$ if all symbols in $xg(x)$ appear in the vocabulary of $T$ and $|xg(x)|$ is bounded by the maximal context window, and if, when $T$ is run on $xg(x)$, its output at position $|x| + i - 1$ is a one-hot vector with "one" at the index corresponding to $g(x)_i \in \Sigma$, for $i = 1, \ldots, |g(x)|$.

We say that the CoT $g(x)$ is *expressible in UHAT* across input lengths if, for each input length $n$, there is a UHAT transformer $T_n$ computing $g(x)$ for all $|xg(x)| = n$, and the numbers of layers and heads are uniformly bounded.

While we bound layers and heads, we do not expect the width to necessarily stay bounded, which allows positional encodings to keep all unboundedly many positions distinct. This is a very weak requirement, a necessary precondition for the existence of a single length-generalizing transformer across input lengths, and much more relaxed than the degree of uniformity often assumed in lower bounds for transformers (e.g. Merrill & Sabharwal, 2023a; Huang et al., 2024). It is nonetheless *sufficient for proving essentially matching lower and upper bounds*.

## 3. Results: Generic CoT Bounds

Our goal is to study the required length of the CoT $|g(x)|$ as a function of the input length $|x|$, under the constraint that the CoT is expressible in the sense of Def. 2.2. Following standard practice in computational complexity, we focus on the worst-case complexity, which – for a given input length $N$, amounts to $\max_{x:|x|=N} |g(x)|$.

It is well-known that transformer CoTs are universal in the sense that they can simulate Turing machines (Pérez et al., 2019; Bhattamishra et al., 2020b; Hou et al., 2024; Merrill & Sabharwal, 2024a; Malach, 2024; Wei et al., 2022a; Qiu et al., 2024). Constructions use a variety of assumptions about attention; we first note that this property holds in our setup:

**Fact 3.1** (Universality of UHAT CoTs)**.** *Consider a Turing machine that terminates within $\leq \tau(N)$ steps on all inputs*

*of length $\leq N$. Then there is a CoT $g(x)$ over a countable alphabet $\Xi$ computing the output of the Turing machine, with length $|g(x)| = \mathcal{O}(\tau(|x|))$, expressible in UHAT.*

The construction is discussed in Appendix B.3. As a consequence, all PTIME problems have polynomial-length CoTs, and problems outside of PTIME cannot have polynomial-length CoTs (Merrill & Sabharwal, 2024a). More generally, CoT is upper-bounded by Turing machine time complexity. The converse does not hold: Many problems have efficient parallel algorithms that can be expressed by self-attention without any CoT. As a simple example, the Boolean AND function of $N$ variables requires $\Omega(N)$ steps with a Turing machine, simply because every input needs to be considered in the worst case, but it can be represented by a very simple transformer with just one attention head, and no CoT. Our main technical contribution in this paper is to show that a diverse set of algorithmic problems *do* require long CoTs, of length linear in the input.

Our key technical tool, and first central result, is a generic CoT lower bound (Theorem 3.3). In order to state and prove this bound, we use the method of *random restrictions*, a key technique for understanding bounded-depth circuits (e.g. Hastad et al., 1994; Furst et al., 1984; Boppana, 1997) and transformers (Hahn, 2020).

**Definition 3.2.** Let $N$ be an input length. A *restriction* $\rho$ is a a family of maps $\rho_N : [1, N] \rightarrow \Sigma \cup \{*\}$, for $N = 1, 2, \ldots$. We write $\rho\Sigma^*$ for the set of strings $x \in \Sigma^*$ where $x_i = \rho_{|x|}(i)$ whenever $\rho_{|x|}(i) \neq *$.

We show that the existence of a sublinear-length CoT has strong implications for the function computed.

**Theorem 3.3** (Generic CoT Bound). *Assume that $f$ has a UHAT-expressible CoT $g(x)$ of length $|g(x)| = o(|x|)$. Choose any $C \in (0, 1)$. Then there is a restriction $\rho$ such that*

1. *$|\{i \leq N : \rho_N(i) = *\}| \geq CN$ for sufficiently large $N$*

2. *For each $N$, $f$ is constant on $\Sigma^N \cap \rho\Sigma^*$*

The first condition says that $\rho$ leaves a large fraction of input positions open; the second condition says that it is nonetheless sufficient for fixing the output.

A simple example of a UHAT-computable function is the AND function, where fixing a single bit to 0 fixes the output to 0; it is easily computed by a UHAT transformer where a single attention head attends to an occurrence of 0 if any exists. Even though every bit can potentially matter to the output, this is easily computed in UHAT without CoT. On the other hand, the PARITY function cannot be fixed by fixing any strict subset of the input bits; hence, Theorem 3.3 entails a linear-length lower bound for a CoT there.

Theorem 3.3 is shown via Lemma 3.4, an analogue of the classical Switching Lemma (Hastad et al., 1994) for $AC^0$ circuits, which permits collapsing bounded-depth circuits into shallow circuits on $\rho\Sigma^*$. Like that classical lemma, it is based on randomizing $\rho$ and showing that the probability of satisfying the desired properties is nonzero; hence, a satisfying $\rho$ exists.

**Lemma 3.4.** *Given a UHAT transformer $T$ operating over the finite alphabet $\Sigma$, and $C \in (0, 1)$, there is $c \in \mathbb{N}$, $K \in \mathbb{N}$ (c and K each only depending on the number of layers and heads, and $|\Sigma|$), and a restriction $\rho$ such that*

1. *$|\{i \leq N : \rho_N(i) = *\}| \geq CN$ if $N \geq K$*

2. *On $\Sigma^N \cap \rho\Sigma^*$, each activation $\boldsymbol{y}_i^{(l)}$ ($i \in [1, N]$, $l \in [1, L]$) depends only on at most $c$ input positions*

The lemma is a strengthening of Theorem 1 of Hahn (2020), the proof is in Appendix B.1. Note that Lemma 3.4 applies to transformer computations without a CoT, whereas Theorem 3.3 broadens the scope to functions computable with a $o(N)$-length CoT. In order to deduce Theorem 3.3, we first apply Lemma 3.4 to obtain a restriction on the input $x$ itself (not the CoT tokens). We then show that it is possible to fix $c \cdot H \cdot L \cdot |g(x)|$ further input symbols to fix *every* symbol appearing in the CoT, and hence the function output $f(x)$. If $|g(x)| = o(|x|)$, the resulting restriction still leaves a constant fraction of input positions free (say, $\frac{9}{10}CN$ positions). We provide the full argument in Appendix B.1.

# 4. Results: Application to Algorithmic Problems

We now proceed to proving explicit lower bounds for various algorithmic problems. For each problem, we first examine sensitivity to establish difficulty for transformers based on the results reviewed in Section 2.1 (independent of unproven conjectures about $TC^0$). For high-sensitivity problems, we then use Theorem 3.3 to lower-bound CoT length, and construct an explicit CoT that matches the bound up to logarithmic factors.

### 4.1. Lower Bounds for Regular Languages

**Definition 4.1.** PARITY takes an input $x \in \{0, 1\}^N$, and decides if the number of 1's is even or odd.

PARITY is the archetype of a highly-sensitive function, with $as_n(\text{PARITY}) = n$, and it has long been documented empirically that transformers struggle with it (e.g. Bhattamishra et al., 2020a; Delétang et al., 2023; Anil et al., 2022; Butoi et al., 2024): We provide the following lower bound:

**Theorem 4.2.** *Any UHAT CoT for PARITY has length $\Omega(N)$.*

The proof is a direct consequence of Theorem 3.3: as long as one does not fix *all* input symbols, one can never fix the output of PARITY. The UHAT bound is tight, and is attained by the straightforward CoT consisting of the parities of increasing prefixes of the input.

We note that Theorem 4.2 is substantively different from bounds based on learnability arguments applying to *subset parities* (e.g. Kim & Suzuki, 2024; Abbé et al., 2024; Abbe et al., 2023; Wies et al., 2022), distinct from the PARITY function applying to the full input (see Section 5 for discussion).

**Why is Theorem 4.2 nontrivial?** At first sight, one might wonder if Theorem 4.2 is trivial: Every bit matters for PARITY, and a hard attention head can only attend to a single position; hence, one might be tempted to argue that a linear number of CoT steps is trivially needed to take every bit into consideration. However, importantly, the behavior of the attention heads itself is input-dependent and can in principle be influenced by every input bit. While each head in UHAT attends to only one position, the entire input globally determines which position this is (e.g., for the boolean AND function of $N$ bits, even though every bit can matter, a single head is enough). Key to showing Theorem 4.2 is that it is sufficient to fix a fraction of input bits to "distract" all attention heads and prevent them from considering the full input. More broadly, our results establish a dichotomy on simulating finite-state automata:

**Corollary 4.3.** *Let $f$ be the membership problem of a finite-state language $L$. Then exactly one of the following holds:*

*1. $L \in AC^0$ and $f$ is expressible in UHAT without CoT*

*2. $L \notin AC^0$, and any UHAT CoT for $f$ has length $\Omega(N)$*

The proof is in Appendix C.1.1. Importantly, $f$ is affected by the sensitivity-based barrier discussed in Section 2.1 if and only $L \notin AC^0$ (Remark C.2). Overall, this result complements results on transformer shortcuts to automata (Liu et al., 2023) by establishing when a shortcut (i.e., a solution with CoT) can be represented well by transformers.

**Experiments** We trained transformers on PARITY on lengths between 1 and 500 (Figure 2 and Appendix C.1.2; results averaged across three runs). We considered the full CoT from Figure 1 (dots along diagonal line). We also considered CoTs where only every $k$-th step of the CoT was provided, shortening the CoT by $1/k$ (dots below diagonal line). Across input lengths, the model is successful when $k \lesssim 3$. We also verified that two LLMs (DeepSeek-R1, Guo et al. (2025), and o1-mini, Jaech et al. (2024)) produce CoTs of at least linear length (Figure 6).

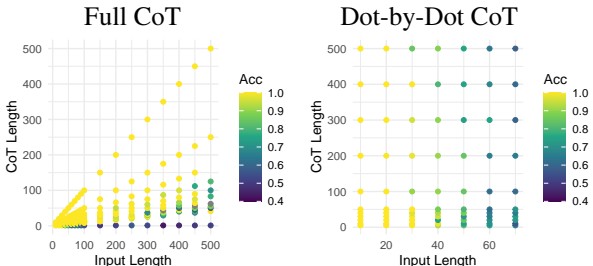

Figure 2: Results for PARITY. Left: CoTs consisting of prefix sum parities, accuracy as a function of input and CoT lengths. CoTs of length $\Theta(N)$ succeed even on very long inputs. Right: Dot-by-dot scratchpads help much less, even when much longer then the input (Theorem 4.11).

## 4.2. Multiplication

We next examine basic arithmetic operations. For $N$-digit numbers, both addition and multiplication are in $TC^0$, and both operations can in principle be expressed by soft-attention transformers (Feng et al., 2024), but empirical research has found transformers to succeed much better at addition than at multiplication, which remains hard for transformers (Yang et al., 2023). Addition has low sensitivity and can be represented in UHAT and fixed precision (Feng et al., 2024); multiplication is shown expressible by unbounded-precision transformers but not fixed-precisioin transformers (Feng et al., 2024). We examine the difficulty of computing each of the $2N$ output digits in multiplication.

**Definition 4.4.** Given two $N$-digit integers $X, Y$ encoded in binary, let $M_k$ be the $k$-th bit of the binary representation of the product $XY$.

Multiplication has high sensitivity; most importantly and interestingly, it turns out that digits in the middle of the result are particularly sensitive (Figure 7). For these:

**Theorem 4.5.** *A UHAT CoT for $M_N$ requires length $\Omega(N)$.*

The proof is in Appendix C.2.1. Importantly, addition faces no such hurdle, as any digit has at most polylogarithmic average sensitivity due to the existence of an $AC^0$ (and UHAT) construction for adding $N$-digit numbers without CoT (Feng et al., 2024); thus, transformers can output each digit in parallel better for addition than for multiplication (Figure 12).

Given our $\Omega(N)$ lower bound (Theorem 4.5), what is the best upper bound? Existing work providing scratchpads for transformers performing multiplication (Hou et al., 2024) uses the naive grade-school algorithm, which has $\Theta(N^2)$ complexity, much larger than the $\Omega(N)$ lower bound. In fact, there is a UHAT scratchpad matching the $\Omega(N)$ lower bound up to a logarithmic factor:

**Theorem 4.6.** *There is a UHAT scratchpad for the full*

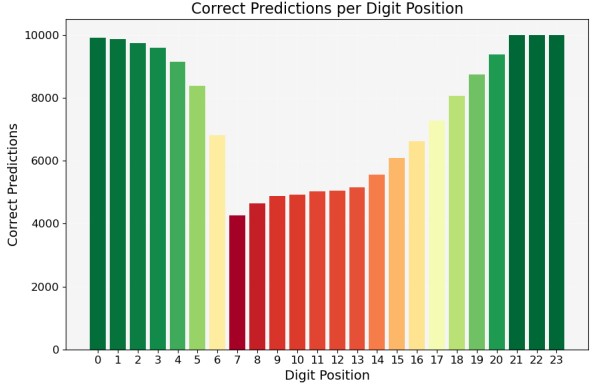

Figure 3: Accuracy of a transformer trained on autoregressive 12-digit multiplication (most significant digit at the left, zero-padded on the left), on a test set with 10K number pairs. High-sensitivity digits in the middle show substantially decreased accuracy even if digits at beginning and end are predicted exactly. See Figure 8 for more.

*product, $M_1 \ldots M_{2N}$, with length $\Theta(N \log N)$.*

The construction is based on the Fourier transform (Appendix C.2.3).

**Remarks**  Theorem 4.5 concerns direct decoding of an individual digit $M_N$; we comment on barriers on *autoregressive* decoding of products in Appendix C.2.2. Recent work has found that custom positional encodings which match structurally corresponding digits to help with addition (Zhou et al., 2023; McLeish et al., 2024; Cho et al., 2024a;b; Sabbaghi et al., 2024). Theorem 4.5 holds for arbitrary positional encoding vectors, suggesting that, while beneficial in the case of ADDITION, specialized positional encodings are insufficient to overcome the difficulty of MULTIPLICATION.

**Experiments**  We trained small transformers (2 layers, 2 heads) to multiply binary numbers with up to 16 digits, either directly, or with the $\mathcal{O}(N \log N)$ CoT from Theorem 4.6. Multiplication failed to generalize when no CoT was given (Figure 12), with difficulty driven by the high-sensitivity middle digits (Figure 3). The CoT succeeded at all lengths up to 16 (Table 1).[4] See Appendix C.2.4 for details. We also verified that two LLMs (DeepSeek-R1 (Guo et al., 2025) and o1-mini (Jaech et al., 2024)) produce CoTs of at least linear length (Figure 6).

### 4.3. Order Statistics

We next consider another task in $TC^0$, which concerns finding order statistics.

---

[4]We focused extensive experiments to numbers of up this length due to resource constraints.

**Definition 4.7.**  Consider the MEDIAN task, where the input consists of $N$ numbers, each with $B$ bits, and the target is the $\lfloor N/2 \rfloor$-th number when the list is sorted.

$TC^0$ circuits solve this by selecting the number that is simultaneously smaller than and greater than $\lfloor N/2 \rfloor$ other numbers. However, the median, especially its last bit, is sensitive to alterations of the $N$ integers (Appendix C.3). We obtain a CoT bound:

**Theorem 4.8.**  *For the MEDIAN task, a UHAT scratchpad requires length $\Omega(N)$. This bound is attained.*

The proof is in Appendix C.3; the bound is attained by a CoT enumerating the lowest $\lfloor N/2 \rfloor$ numbers.

**Experiments**  We trained transformers on MEDIAN on input lengths between 6 and 398 (Figure 5 and Appendix C.3.2), corresponding to 1 to 99 decimal numbers (each with $B$ digits). We considered the full CoT (upper most dots in each input length). Like PARITY, we also considered CoTs where only every $k$-th step of the CoT was provided (starting from the smallest number), shortening the CoT by $1/k$ (dots below the upper most dots in each input length). Across input lengths, the model is always successful when using the full CoT, and there is a chance of failure when $k \geq 2$ (accuracies are averaged over 3 runs).

### 4.4. Graph Reachability

Another foundational aspect of reasoning is graph reachability.

**Definition 4.9.**  Given a set of vertices $V$ and a set of directed edges $E \subseteq V \times V$, the Reachability task is defined as follows. *Input*: Given some numbering of $V$, we assume a list of all edges $E$, each coded as a pair of $\log V$-digit binary numbers. We also assume a query pair of two vertices $i, j$. *Output*: A binary label, indicating if there is a path from $i$ to $j$.

The reachability problem in graphs, even in directed acyclic graphs (DAG), is **NL**-complete (Jones et al., 1976); hence, by the unproven conjecture $TC^0 \neq NC^1$, is expected not to be solved by transformers without CoTs. We provide an *unconditional lower bound*, even for a sub-family of graphs for which the reachability problem is solvable in $TC^0$, by observing that PARITY is reducible to DAG reachability:

**Theorem 4.10.**  *There is a family $\mathfrak{G}$ of DAGs inside which reachability is solvable in $TC^0$, but cannot be represented by a transformer at sublinear average sensitivity. A UHAT CoT needs length $\Omega(|E| \log |V|)$. This bound is attained.*

The proof proceeds by coding PARITY into DAGs with $2N$ vertices (lower bound), and coding breadh-first search into a CoT (upper bound), see Appendix C.4.

**Experiments** We generate random DAGs with sizes ranging from 5 to 35 vertices and use them to train Transformers to solve a DAG reachability task in the general case with a $\mathcal{O}(|E|\log|V|)$-length scratchpad. The results, presented in Figure 4, show that a scratchpad of this length is indeed sufficient to solve the task. In contrast, a model trained without a scratchpad performs at chance level. See Appendix C.4.2 for details.

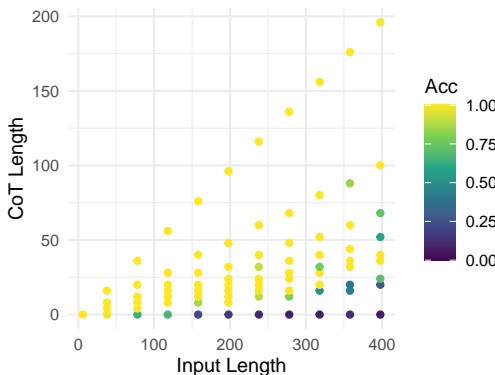

Figure 5: Results for MEDIAN: CoTs consisting of the sorted first $\lfloor N/2 \rfloor$ numbers. Accuracy as a function of input and CoT lengths.

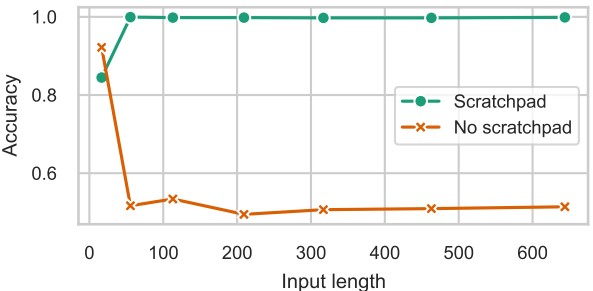

Figure 4: Results for DAG reachability. Without a CoT, the model performs at chance level for all but very small graphs. An $\mathcal{O}(|E|\log|V|)$-length CoT leads to a perfect accuracy independent of the graph size.

### 4.5. Limitations of Dot-by-Dot CoTs

Recent work has observed a benefit even for CoTs consisting just of repetition of a single token ("pause token"), which might give a transformer the opportunity to perform extra computations even without outputting any intermediate steps (Goyal et al., 2024; Pfau et al., 2024). While the power of such "dot-by-dot" CoTs remains in $TC^0$, provided their length is polynomially bounded, they may still enable additional computations (Pfau et al., 2024), which raises the potential challenge of unauditable unobservable computations in LLMs. Our methods result in additional barriers for such CoTs, suggesting that high-sensitivity computations require a substantial degree of reliance on *explicit* CoT tokens. Specifically, for PARITY, such a CoT cannot have polynomially bounded length:

**Theorem 4.11.** *Consider a UHAT-expressible CoT for PARITY where $g(x)$ has the form $\ldots\ldots\#f(x)$. ("dot-by-dot CoT"). Then $|g(x)| = \omega(|x|^k)$ for any $k \in \mathbb{N}$.*

The proof is in Appendix B.2. Complementing this lower bound, an exponentially-sized UHAT CoT does exist in principle (Appendix B.2). Overall, we have thus a super-polynomial separation between the lengths required for full CoTs ($\Theta(N)$) and dot-by-dot CoTs ($\omega(poly(N))$)

**Experiments** Figure 2B shows that a dot-by-dot variant of the CoT for PARITY is not successful at expanding the

lengths at which a transformer is learned successfully.

## 5. Discussion

### 5.1. Implications

Across Sections 4.1–4.4, our lower bounds are tight up to polylogarithmic factors. By showing that the UHAT CoT length must scale with the input length, they provide barriers on the possibility for self-attention to solve these tasks with small inference-time compute in generating additional tokens.

An attractive approach for shortening CoT reasoning and reducing inference-time compute is by fine-tuning transformers to perform reasoning in fewer and fewer steps (Deng et al., 2024). Our results show that such a process is likely to run up against representational limitations of the architecture. The findings demonstrate efficiency limitations to CoT-based reasoning that might be overcome via tool use or stronger architectures. Logarithmic growth of the depth may be one way to overcome these (Merrill & Sabharwal, 2024b); within fixed-depth architectures, certain SSM architectures overcome at least the difficulty of PARITY (Grazzi et al., 2024). It remains to be understood if this applies to the other problems from Sections 4.1–4.4.

### 5.2. Related Work

**Theoretical Understanding of Scratchpads and CoT** The first theoretical study of scratchpads for transformers is by Pérez et al. (2019), who showed that transformers can autoregressively simulate the computations of a Turing machine. More recently, work has shown that polynomial-length CoT can allow transformers to transcend the complexity class $TC^0$ (Feng et al., 2023; Merrill & Sabharwal, 2024a; Li et al., 2024), and express all problems in PTIME,

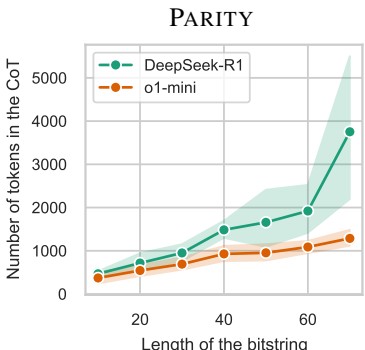 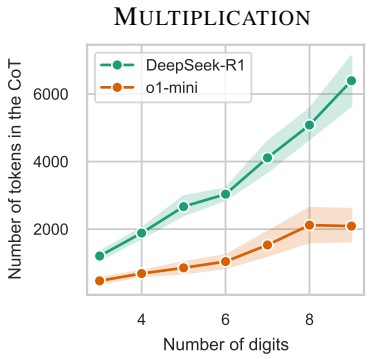 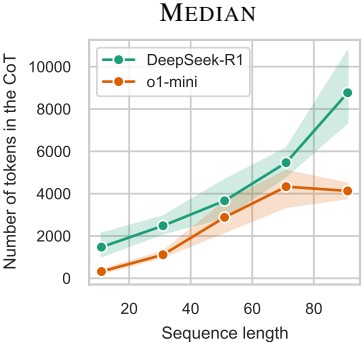

Figure 6: Length of reasoning traces of DeepSeek R1 (Guo et al., 2025) and o1-mini (Jaech et al., 2024) LLMs prompted to solve the tasks of parity (left), multiplication (center), and median (right). We only include traces leading to the correct answer. In both cases, the growth of CoT appears at least linear. Manual inspection of reasoning traces of DeepSeek R1 shows that the model perform the counting of all ones for PARITY, and the naive quadratic algorithm for MULTIPLICATION. The experimental details are provided in Appendix D.

which by widely believed but unproven conjectures far exceeds $TC^0$. There are also lower bounds for scratchpads for *single-layer* transformers (Peng et al., 2024; Barceló et al., 2025), showing that a one-layer transformer (both the input itself and the CoT are processed by a single layer) requires a substantial number of steps to solve certain problems, such as iterated function composition. Chen et al. (2024) extended this line of work to multi-layer transformers, proving a benefit for CoT in iterated function composition.

Another angle to understanding CoTs is via learnability arguments (Wies et al., 2022; Kim & Suzuki, 2024; Hahn & Goyal, 2023; Abbé et al., 2024). A particularly promising angle is the *globality degree* (Abbé et al., 2024), with potentially broad implications for scratchpads, though resulting lower-bounds remain largely conjectural beyond special cases (Appendix E.2). Interestingly, the globality degree is linked to our sensitivity-based techniques and could in principle lead to stronger bounds; a future full proof of the main conjecture of Abbé et al. (2024) would, in combination with our unconditional results here, entail even further results (Appendix E.2).

**Lower bounds for transformers** Lower bounds for UHAT transformers primarily rest on random restrictions, either directly (Hahn, 2020; Barceló et al., 2024) or via reduction to known circuit bounds proven with those (Hao et al., 2022). A key technical step in our work is to expand the reach of the random restriction technique to transformers including CoTs. Another approach to obtaining lower bounds relies on circuit conjectures from computational complexity (Merrill & Sabharwal, 2023b; Sanford et al., 2024a) and is thus conditional on these (widely believed) conjectures. There are further techniques for single-layer transformers (Kozachinskiy, 2024; Peng et al., 2024; Sanford et al., 2024b; Bhattamishra et al., 2024; Barceló et al.,

2025), for autoregressive transformers (Chen et al., 2024), and for length-generalizng transformers (Huang et al., 2024) relying on communication complexity or VC dimension.

**Relation to Results on Subset Parities** Our results concerning PARITY are distinct from results on *Subset Parities*. A long string of results has established difficulty of learning subset parities – that is, parity functions restricted to subsets $U \subseteq \{1, \ldots, N\}$ – in various setups, and established benefits for providing intermediate steps (e.g. Wies et al., 2022; Abbé et al., 2023; Kim & Suzuki, 2024; Yang et al., 2025). Intuitively, the difficulty of learning subset parities emerges from the fact that one has to identify the set $U$ from the exponentially-large power-set. In learning such functions, there is a provable benefit to providing intermediate steps in training (Wies et al., 2022; Abbé et al., 2023; Kim & Suzuki, 2024; Yang et al., 2025). However, this argument does not readily imply that causal transformers will find the PARITY function difficult to learn and that CoTs will help there, because the PARITY function applies to a very specific set ($U = \{1, \ldots, N\}$). Our results thus are distinct from results on subset parities due to Wies et al. (2022); Abbé et al. (2023); Kim & Suzuki (2024); Yang et al. (2025). Interestingly, Abbé et al. (2024) show that PARITY may be easy to learn for MLPs with Rademacher initialization; however, this requires a weight norm substantially growing with $N$, making transformer length generalization unlikely (Huang et al., 2024).

**Mechanistic Understanding of CoT** Concurrent with the theoretical line of work, the mechanisms behind the CoT abilities in LLMs have been studied empirically. Wang et al. (2024) used path patching to identify attention heads critical for CoT and found that only a few attention heads are important for that. Dutta et al. (2024) showed that LLMs

employ multiple neural pathways that work in parallel to generate the final answer, all writing to the residual stream. Kudo et al. (2024) demonstrated for arithmetic reasoning that single-step subproblems can be solved by LLMs even before the scratchpad begins, while more complicated tasks are solved during CoT generation.

### 5.3. Limitations and Open Questions

One limitation of our bounds is that they apply in the worst-case setting. Providing average-case bounds is an interesting problem for future research. A second limitation of our results is that they focus on the UHAT model, while there are other formal models of transformers (e.g. log-precision transformers, Merrill & Sabharwal (2023a)). Our results pose the open question whether some other model might enable substantially shorter but still learnable CoTs:

**Open Problem 5.1.** *Are there sublinear-length CoTs for any high-sensitivity problem from Sections 4.1–4.4 that are expressible by some formal model of transformers (other than UHAT; e.g., log-precision transformers), while being practically learnable?*

Solving this problem in the affirmative might have substantial implications for real-world LLMs, and any such solution is likely to require further progress in understanding the learnability properties of transformers. On the other hand, a negative answer might suggest that tool use or architectural improvements are likely to be unavoidable for transformer-based LLMs even on many $TC^0$ problems, due to architectural limitations that make successful CoTs impractically long.

## 6. Conclusion

The success of scratchpad and chain-of-thought techniques raises the need to understand their strengths and limitations. Here, we have provided lower bounds for the number of steps required for CoT reasoning in various fundamental algorithmic problems, in the UHAT model. These bounds are tight up to polylogarithmic factors, and are attained by realistically trainable models. Taken together, our results provide theoretical bounds on the ability of transformers to solve reasoning problems with CoT.

## Impact Statement

This paper presents work whose goal is to advance the field of Machine Learning. There are many potential societal consequences of our work, none which we feel must be specifically highlighted here.

## Author Contributions

MH coordinated the project. All authors contributed to the conceptual framework. Experiments were contributed by AA (Section 4.2), XH (Section 4.3), MR (Section 4.4), and MH (Sections 3, 4.1, and 4.5). MH contributed the bounds for UHAT transformers, with input from the other authors. AA and MH jointly developed theory in Section 4.2.

## Acknowledgments

Funded by the Deutsche Forschungsgemeinschaft (DFG, German Research Foundation) – Project-ID 232722074 – SFB 1102. The authors thank Satwik Bhattamishra, Will Merrill, Sophie Hao, Anej Svete, Franz Novak, Ryan Cotterell, Emmanuel Abbé, and Andy Yang for discussion, and Yash Sarrof and Entang Wang for feedback on the paper.

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

# Appendix Contents

## A. FAQ

1. *Isn't it obvious that one needs at least a linear number of steps to solve these algorithmic problems?*

   In general, serial computation models such as Turing machines need at least a linear number of steps in order to solve tasks that require knowing the full input. The situation is different for parallel computation as performed by transformers: Many problems do have direct parallel solutions. For instance, for all regular languages expressible in the circuit complexity class $AC^0$, transformers can express these in the UHAT model without CoT (Theorem 4.3), essentially via parallel 'shortcuts'. Our results are notable in that they establish that many algorithmic problems do require linear-length CoTs, with barriers on the possibility of substantial parallel shortcuts.

2. *All lower bounds in this paper are essentially on the order of $\Omega(N)$ or $\Omega(N \log N)$. Do any algorithmic problems require substantially super-linear (e.g., quadratic) CoTs?*

   This question is closely linked to deep unsolved questions at the heart of computational complexity. Due to the Turing completeness of transformer CoTs, proving such lower bounds on CoTs would entail superlinear (e.g., quadratic) lower bounds on multitape Turing machine time complexity, which has been extremely challenging, even for NP-complete problems.

3. *Transformers are already known to be Turing-complete. Why does this paper construct CoTs for the algorithmic problems – isn't reducing to known Turing machine constructions enough?*

   Compared to modern random-access models, single-tape Turing machines (as used in typical Turing machine completeness proofs for transformers) require substantial overhead to process data structures central to many algorithmic problems. For instance, a straightforward implementation of the queue used in breadth-first search (Theorem 4.10) takes a quadratic number of steps on a single-tape Turing machine, as the head needs to repeatedly move between start and end position of the queue. It is thus not immediate from Turing completeness that these problems can all be solved at (near-)linear CoT lengths. Multi-tape Turing machines (which can also be coded into CoTs) may provide substantially better bounds, but CoTs derived from Turing machine constructions are still likely to have substantial overhead (even if just constant factors). In contrast, we show that explicit constructions on the four algorithmic problems studied can be practically learned, confirming that the upper bounds are meaningful.

4. *Implementations of self-attention already have quadratic complexity. Why should one be concerned with a further linear number of CoT steps?*

   It is true that generating $N$ tokens, with KV-Cache, will have just $\mathcal{O}(N^2)$ complexity, asymptotically the same as directly providing an answer. However, it can still lead to substantial overhead, even if just by a constant factor, if the number of CoT tokens grows appreciably with $N$ (compare Figure 6). Thus, any potential sublinear CoTs would be of great interest in increasing efficiency; our results demonstrate barriers to such solutions.

5. *What is the practical impact of the results? Do the results relate to any specific NLP tasks?*

   PARITY, MULTIPLICATION, MEDIAN, REACHABILITY are foundational problems, which instantiate simple models of reasoning problems that have been of broad interest. For instance, PARITY is a simple example of state tracking, a family of reasoning problems that have been of substantial interests and that still pose challenges for LLMs (e.g. Merrill et al., 2024; Kim & Schuster, 2023; Hsieh et al., 2024). There also has been a large amount of interest in the ability of transformers and LLMs to perform arithmetic such as MULTIPLICATION, as it is a fundamental ingredient of mathematical reasoning. REACHABILITY is a simple case of search, which is foundational to many aspects of reasoning, and transformers' abilities to perform such problems have been an object of interest (e.g. Lehnert et al., 2024). On all these problems, our results entail barriers on the possibility of transformer algorithms avoiding substantial CoTs.

6. *Definition 2.1 enforces a finite input alphabet, but allows an infinite CoT alphabet. Why?*

   The input alphabet needs to be finite for Theorem 3.3 to go through, because the proof of Lemma B.3 by Hahn (2020) involves a union bound over the alphabet. On the other hand, this is not needed for the CoT tokens. Some prior work (Bhattamishra et al., 2020b; Abbé et al., 2024) allows the CoT vocabulary to grow with the input length. Theorem 3.3 can accommodate this, and we indeed find this useful in our constructions.

7. *Definition 2.2 allows using different transformers $T$ at every input length. Isn't this unrealistic? What about the role of length generalization?*

   It is true that, in practice, one will expect transformers to perform the tasks across input lengths. However, the theoretical literature on transformers has developed different approaches to formalize this. Notably, a single transformer at fixed width may have trouble solving a single task across unboundedly many input lengths unless very specific positional encodings are used (e.g. Merrill & Sabharwal, 2024a), and work has thus often considered cases where the width may grow with the length (e.g. Liu et al., 2023; Huang et al., 2024; Bhattamishra et al., 2024). To accommodate such variability, we aim for the most general setup under which we can prove lower bounds. We hence expect only the depth and the number of heads to stay constant, but allow everything else to grow with the input length. This is the most general setup under which our techniques allow us to show Theorem 3.3.

8. *The results in this paper are proven for a hard-attention model, whereas real-world implementations use softmax attention, which can express functions that hard-attention transformers cannot. In particular, UHAT is bounded by $AC^0$, whereas softmax transformers are bounded by $TC^0$. Given this difference, what do the results entail about real-world LLMs?*

   It is true that softmax attention can express some functions that hard attention (and $AC^0$) cannot, such as the MAJORITY function. On the other hand, $TC^0$ is an overly optimistic upper bound on practical abilities of transformers, as many $TC^0$ functions are not practically learned by transformers due to the sensitivity limitations discussed in Section 2.1. Hence, we focused on tasks for which existing results based on average sensitivity (Hahn & Rofin, 2024) entail that transformers will struggle across different formal models of self-attention, irrespective of differences in expressive capacity (Section 2.1), including problems within $TC^0$. Thus, for the tasks considered here, a CoT is practically needed irrespective of the specific formalization of self-attention; this is also confirmed by our experiments, where transformers consistently failed in the absence of CoTs. We further note that existing CoT constructions from the literature can by and large be expressed well using hard-attention operations (Appendices E.1 and B.3). In LLM experiments, we observed algorithms in line with our theoretical constructions (Appendix D). Our results thus place strong constraints on any possibility of evading linear lower bounds. In principle, it is possible that other formal models of transformers allow substantially faster CoTs on some algorithmic problems while maintaining practical learnability (Problem 5.1); rigorously proving or refuting the existence of such solutions is likely to require substantial advances in understanding transformers' learnability properties.

9. *CoT lower bounds are proven for hard attention, while experiments are conducted with softmax attention. Isn't there a mismatch between theory (hard attention) and experiments (softmax transformers)?*

   It is true that we do not conduct experiments in the hard-attention setup, because there is no efficient training procedure for that setup. Our experiments, conducted in the practical softmax attention setting, complement the theoretical results in two ways:

   (a) First, transformers consistently failed in the absence of CoT on the four algorithmic tasks. This supports the theoretical prediction that a CoT improves transformers' ability to solve these tasks.

   (b) Second, by showing that CoTs implementing the theoretical upper bounds can be practically learned, we demonstrate the real-world meaningfulness of our theoretical bounds.

## B. General Theoretical Results

### B.1. Proof of Theorem 3.3: Generic CoT Bound (Main Result)

Here, we prove our main lower bound (Theorem 3.3), which provides a generic condition under which sublinear CoTs can be possible. Our techniques build on lower-bounding methods for bounded-depth circuits (Furst et al., 1984; Hastad et al., 1994). Recall the definition of restrictions (Definition 3.2).

**Definition B.1.** We say $\rho' \succ \rho$ if, whenever $\rho'_N(i) = *$, then $\rho_N(i) = *$ ($1 \leq i \leq N$).

**Definition B.2** (*c*-Transformer). A *c*-Transformer conforms to the definition of transformers from Section 2.2, except that each $\boldsymbol{y}_i^{(0)}$ is a function of $\leq c$ input positions. That is, there are functions $f_i$ such that:

$$\boldsymbol{y}_i^{(0)} = f_i(x_{1...N}) \tag{6}$$

and $f_i$ depends on at most $c$ of its inputs.

We build on the following lemma shown by Hahn (2020), rephrased for self-containedness. It can be viewed as a transformers analogue of the Switching Lemma (Hastad et al., 1994) for bounded-depth circuits:

**Lemma B.3** (Depth Reduction Lemma). *Given a UHAT $c$-Transformer $T$ with $L$ layers, a constant $C \in (0,1]$, and restrictions $\rho_N = \rho_1, \rho_2, \dots$ such that*

$$|\{i \leq N : \rho_N(i) = *\}| \geq CN \tag{7}$$

*for all sufficiently large $N$.*

*Choose any $C' \in (0, C)$. Then there are restrictions $\rho_1' \succ \rho_1, \rho_2' \succ \rho_2, \dots$ such that*

$$|\{i \leq N : \rho_N'(i) = *\}| \geq C'N \tag{8}$$

*for all sufficiently large $N$, and such that there is a $c \cdot (|\Sigma|^c kH + 1)$-transformer $T'$ with $L - 1$ layers, for some integer $k$ (depending on $C'$), such that, for each $i \in [1, N]$ and each $l \in [1, L]$,*

$$\boldsymbol{y}_i^{(l)} \text{ in } T$$

*equals*

$$\boldsymbol{y}_i^{(l-1)} \text{ in } T'$$

*whenever the input is in $\rho' \Sigma^*$.*

*Proof.* This is a stronger statement of Lemma 4 of Hahn (2020). That lemma only stated that the final output $\boldsymbol{y}_N^{(L)}$ was preserved in $T'$. However, by the way $T'$ is obtained in that proof[5], the stronger statement for all activations follows. We also note that the original statement considered $\Sigma = \{0, 1\}$, but the proof transfers to arbitrary finite alphabets $\Sigma$ without change, except that $T'$ is now a $c \cdot (|\Sigma|^c kH + 1)$-transformer, rather than a $c \cdot (2^c kH + 1)$-transformer.

We note that Hahn (2020) did not assume causal masking, whereas we are assuming it by default (Section 2.2). Causal masking can be easily simulated in UHAT, by setting up positional encodings in such a way that $a_{i,j}^{(k,h)}$ is extremely small whenever $j > i$. Hence, the proof of the Depth Reduction Lemma continues to work in the presence of hard-coded causal masking. $\square$

This lemma entails the following fact, a strengthening of Theorem 1 of Hahn (2020):

**Lemma B.4** (Restated from Lemma 3.4). *Let $T$ be a UHAT transformer operating over a finite alphabet. Take any $C \in (0, 1)$. Then there is $c \in \mathbb{N}$, $k \in \mathbb{N}$ such that, for each $N > k$, there is $\rho$ such that*

1. *$|\{i : \rho(i) = *\}| \geq CN$*

2. *within $\rho \Sigma^*$, each $\boldsymbol{y}_i^{(l)}$ is determined by $\leq c$ input positions*

*Proof.* By applying Lemma B.3 iteratively, $L$ times. $\square$

*Remark* B.5. A reader might wonder why, instead of introducing Lemma B.4, one cannot simply apply Theorem 1 of Hahn (2020) independently to each layer of a Transformer. The reason is that the latter approach would only ensure the existence of $L$ independent restrictions – one for each layer –whereas Lemma B.4 guarantees that a single restriction fixes each $y_i^{(l)}$.

We now conclude the key result, Theorem 3.3:

**Theorem B.6** (Restated from Theorem 3.3). *Assume that $f$ has a UHAT-expressible CoT $g(x)$ of length $|g(x)| = o(|x|)$. Choose any $C \in (0, 1)$. Then there is a restriction $\rho$ such that*

1. *$|\{i \leq N : \rho_N(i) = *\}| \geq Cn$ for sufficiently large $N$*

---

[5]"We can thus remove layer 0, convert layer-1 activations $\boldsymbol{y}_j^{(1)}$ into layer-0 activations $\boldsymbol{y}_j^{(0)}$, and obtain a $(c \cdot (2^c kH + 1))$-transformer performing the same computation as before when $\rho^{(3)}$ is applied." (quoted from Hahn, 2020). Here, "$\rho^{(3)}$" corresponds to the restriction $\rho'$ from the statement of the lemma.

*2. For each $N \in \mathbb{N}$, $f$ is constant on $\Sigma^N \cap \rho\Sigma^*$*

*Proof.* First, we obtain a restriction $\rho^{(0)}$, and integers $c$, $k$ from Lemma 3.4, applying $T$ only on the input $x \in \Sigma^*$ itself.

For any $N > k$, we now consider the following. Let $M := \max_{|x|=N} |g(x)|$. Without loss of generality, we can pad the CoTs for all $x$ with $|x| = N$ to have length $M$.

Let $g(x) = g_1 \ldots g_M \in \Xi^M$.

We now construct a sequence $\rho^{(0)} \prec \rho^{(1)} \prec \rho^{(2)} \prec \rho^{(3)} \prec \ldots \rho^{(M)}$ while maintaining the following properties for $k = 1 \ldots M$ (†):

1. $\left| \{i : \rho_N^{(k)}(i) = *\} \right| \geq CN - k \cdot c \cdot H \cdot L$

2. For each $l \in [0, L]$, $i \in [1, k-1]$, the activation $\boldsymbol{y}_{N+i}^{(l)}$ is constant on all strings in $\Sigma^N \cap \rho_N^{(k)}\Sigma^*$.

3. $g_1 \ldots g_k \in \Xi^k$ is constant on all strings in $\Sigma^N \cap \rho_N^{(k)}\Sigma^*$.

We prove this by induction.

**Inductive Base** ($k = 1$) We can fix $\boldsymbol{y}_N^{(1)}, \ldots, \boldsymbol{y}_N^{(L)}$ by fixing at most $\leq c \cdot L$ input positions, obtaining $\rho^{(1)} \succ \rho$, with

$$\left| \{i : \rho^{(1)}(i) = *\} \right| \geq CN - c \cdot L \geq CN - 1 \cdot c \cdot H \cdot L \tag{9}$$

Since $g_1$ is determined by $\boldsymbol{y}_N^{(L)}$, this also fixes $g_1$.

**Inductive Step** ($k > 1$) Assume that the claim has been shown for all $k' < k$ (Inductive Hypothesis). Note that $g_k$ is determined by $\boldsymbol{y}_{N+k-1}^{(L)}$. By the Inductive Hypothesis, $g_{k-1}$ is fixed by $\rho^{(k-1)}$; hence, $\boldsymbol{y}_{N+k-1}^{(0)}$ also is. We now perform the following construction for each layer $l = 1, 2, \ldots, L$, and – within each layer – for each head $h = 1, \ldots, H$. We iteratively expand $\rho^{(k-1)}$ by restricting more input tokens.

At a given layer $l$ and head $h$, let $\tilde{\rho} \succ \rho^{(k-1)}$ be the restriction obtained after treating layers $1, \ldots, l-1$ and, within layer $l$, heads $1, \ldots, h-1$. The activation $\boldsymbol{y}_{N+k-1}^{(l-1)}$ is now already fixed by $\tilde{\rho}$. Our goal is now to restrict a few more input positions to fix the attention of this head to a specific position, and thus fix its output. For this, determine the maximal value of any attention score over all inputs satisfying the restriction, and fix input tokens so that this value is achieved, forcing the attention head to attend to a specific position. Formally, for each $j \in [1, N+k-1]$, we consider

$$A_j := \max_{x \in \Sigma^N \cap \rho^{(k-1)}\Sigma^*} a_{N+1,j}^{(l,h)} \tag{10}$$

Now let $\widehat{j}$ be the $j$ that maximizes $A_j$ (under the tie-breaking procedure, e.g. choosing the leftmost one, if more than one attain the same $A_j$). One possibility is that $\widehat{j} \in [N+1, \ldots, N+k-1]$; then attention is guaranteed to fall onto this position because each relevant activation is constant across $\rho^{(k-1)}\Sigma^*$ by Inductive Hypothesis. The more interesting possibility is when $\widehat{j} \in [1, N]$. Note that $\boldsymbol{y}_{\widehat{j}}^{(l-1)}$ depends only on $\leq c$ input tokens in $\rho\Sigma^*$; hence, we can expand the restriction $\tilde{\rho}$ by fixing $\leq c$ additional input symbols to force $\boldsymbol{y}_{\widehat{j}}^{(l-1)}$ to take on a value leading to that maximal attention score $a_{N+k-1,\widehat{j}}^{(l,h)}$. Overall, performing this sequentially on each layer and head, we ultimately fix all activations $\boldsymbol{y}_{N+k-1}^{(l)}$ and hence $g_k$, with a restriction $\rho^{(k)}$ that restricts at most $\leq c \cdot L \cdot H$ further input tokens beyond $\rho^{(k-1)}$.

This concludes the inductive proof of the claim (†). Overall, by taking $\rho := \rho^{(M)}$, we have fixed an additional $\leq c \cdot H \cdot L \cdot |g(x)| = o(N)$ input positions to fix all CoT tokens. As the output $f(x)$ is part of the CoT, $f(x)$ must also be constant across all $x \in \Sigma^N \cap \rho_M\Sigma^*$. $\square$

*Remark* B.7. We assume, by default, causally masked attention throughout the transformer (Section 2.2), for consistency with typical modern language models. Our results are also compatible with setups where attention is bidirectional on the input and causal masking at most applies during CoT generation, which is the setup assumed by Abbé et al. (2024). As explained in the proof of the Depth Reduction Lemma, it holds independently of whether causal masking is applied to the input or not.

### B.2. Proof of Theorem 4.11 (Barriers for Dot-by-Dot Scratchpads)

**Theorem B.8** (Repeated from Theorem 4.11). *Consider a UHAT-expressible CoT for* PARITY *where $g(x)$ has the form*

$$\ldots . . \# f(x) \qquad (\text{``dot-by-dot CoT''}) \tag{11}$$

*This CoT has length $\omega(|x|^k)$ for all $k > 0$.*

*Proof.* In such a CoT, we can consider the suffix $f(x)$ as the target of prediction given the input $x \ldots \#$ of length $n := |x| + |g(x)| - 2$, with $N := |x|$. The function predicting $f(x)$ based on a prefix $x \ldots \#$ acts on an input of length $n = N + |g(x)| - 2$, where $N = |x|$, and has average sensitivity $N$. We note the following fact: Due to the inclusion of UHAT in $AC^0$ (Hao et al., 2022), and the known bound on the average sensitivity of $AC^0$ circuits (Corollary 12.14 in Jukna (2012)); originally due to Linial et al. (1993); Boppana (1997)), we have $as_N(f) = \mathcal{O}(poly(\log(n)))$. Hence, $N = \mathcal{O}(poly(\log(N + |g(x)|)))$; hence $|g(x)|$ cannot be bounded by a polynomial of $N$. □

**Theorem B.9** (Exponentially-Sized Dot-by-Dot Scratchpad for PARITY). PARITY *has a UHAT dot-by-dot CoT of length* $\mathcal{O}(\exp(N))$

*Proof.* We lay out a construction for a dot-by-dot scratchpad of length $2^N + 2$, expressible in a UHAT Transformer with two layers, one attention head, and $d = 2N$ hidden dimensions. Each of the first $2^N$ positions in a scratchpad is assigned a unique bitstring $\xi^{(i)} \in \{0, 1\}^N$ ($i \in [1, 2^N]$): that is, $i$-th token in a scratchpad (or $N + i$-th token overall) corresponds to a binary encoding of $i - 1$.

The construction works as follows. In the first layer, each of the $2^N$ scratchpad tokens assigned with $\xi^{(i)}$ determines whether the input string is equal to $\xi^{(i)}$. In the second layer, the $(2N + 1)$-th scratchpad token gathers information from them and uses it to predict the hard-coded value of parity for the input string (which is the $(2N + 2)$-th scratchpad token).

In the first layer, at the position corresponding to string $\xi^{(i)}$, an attention head sends out a query looking for positions $j$ where $x_j \neq \xi_j^{(i)}$. For this, the query corresponding to bit string $\xi^{(i)}$, emanating from the $i$-th position in the CoT (overall, index $N + i$), has the form

$$\begin{pmatrix} \xi_1^{(i)} & \ldots & \xi_N^{(i)} & (1 - \xi_1^{(i)}) & \ldots & (1 - \xi_N^{(i)}) \end{pmatrix} \tag{12}$$

At this position, the key and value vectors are zero.

The key and value corresponding to the $j$-th bit in the input has the form $K_j$:

$$K_j = \begin{cases} e_j & x_j = 0 \\ e_{N+j} & x_j = 1 \end{cases} \tag{13}$$

where $e_{\ldots} \in \mathbb{R}^{2N}$ are one-hot vectors.

Keys of all tokens in the scratchpad are constant vectors of -1.

Then, the attention score

$$a_{N+i,j} = \begin{cases} 1 & x_j \neq \xi_j^{(i)} \\ 0 & x_j = \xi_j^{(i)} \\ < 0 & j > N \end{cases} \tag{14}$$

If one position $j$ achieves a 1, one of these will be selected by tie-breaking; if none does, one of the $N$ input bits will be selected. The MLP at the query position checks if the retrieved value vector indeed indicates a mismatch $x_j \neq \xi_j^{(i)}$, using knowledge about $\xi^{(i)}$ forwarded via the residual connection. If no mismatch is found, the transformer can conclude that $x = \xi^{(i)}$, and use that in the second layer.

In the second layer, we are interested in the $(2^N + 1)$-th scratchpad token, superceeding all positions corresponding to $\xi^{(i)}$. At this position, the second-layer attention head attends to the unique position at which the first layer found no mismatch, and retrieves the hard-coded answer computed from the positional embedding of that position. □

## B.3. Proof of Fact 3.1 (Universality of UHAT CoTs)

A substantial number of constructions coding Turing machine computations into transformer CoTs have been presented in the literature (e.g. Pérez et al., 2019; Wei et al., 2022a; Bhattamishra et al., 2020b; Hou et al., 2024; Qiu et al., 2024; Malach, 2024), but they all use different assumptions about the formal model of transformers. For self-containedness, we here provide a simple construction. Our construction is largely equivalent to that of Wei et al. (2022a), but we present it in a simplified and self-contained manner in the notation used in our paper. We also note that constructions from Bhattamishra et al. (2020b); Hou et al. (2024); Qiu et al. (2024); Malach (2024) can also be expressed straightforwardly in UHAT. We discuss existing constructions from the literature at the end of this subsection.

**Construction**   Consider a Turing machine defined by

1. a finite tape alphabet $\Sigma$

2. a finite state set $Q$

3. the action set

$$\mathcal{A} := \{\text{LEFT}, \text{RIGHT}\} \cup \{\text{WRITE}(\sigma) : \sigma \in \Sigma\} \tag{15}$$

4. a transition function $\delta$, mapping $\Sigma \times \mathcal{Q}$ to $\mathcal{A} \times \mathcal{Q}$

5. a start state $s_0 \in Q$

6. a terminating set $T \subset Q$

with the following computation:

1. We consider a tape with positions $0, 1, 2, \ldots$.

2. At the beginning, the machine starts at position $0$ and in state $s_0$; the tape holds a finite input word starting at $0$, ending in a separator symbol. All remaining tape positions hold a blank symbol $\text{BLANK} \in \Sigma$.

3. At each step, the next action and state are decided based on $\delta$ applied to the current state and the current tape symbol

4. The machine stops when a state from $T$ is reached

5. The final state indicates whether the input word was accepted or rejected

We encode the computations as follows. We first encode the input word as a string over the input alphabet $\Sigma$, followed by a separator symbol. We then construct a CoT over the infinite alphabet

$$\Xi := \mathbb{N} \times \mathcal{A} \times \mathcal{Q} \tag{16}$$

For each state transition, we record (i) the tape position after carrying out $\delta$ (an element of $\mathbb{N}$), (ii) the output of $\delta$ (an element of $\mathcal{A} \times \mathcal{Q}$).

We relate this construction to other constructions in Appendix E.

We now show that this is implemented by UHAT as defined in Section 2.3. We need to define a set of UHATs $T_N$, operating on inputs with length $\leq N$, with uniformly bounded number of layers and heads. We define

$$\Xi_N := [0, N] \times \mathcal{A} \times \mathcal{Q} \tag{17}$$

First, let $N$ be a bound on the input length. At each position,

1. The input token $\xi = (i, a, q) \in [0, N] \times \mathcal{A} \times \mathcal{Q}$ provides the tape position $i$, the output $a$ of $\delta$, and the resulting state $q$.

2. An attention head attends to the last step at which the Turing machine head had been at tape position $i$ while doing a WRITE operation.

The key, given input token $\xi' = (i', a', q')$, is the one-hot vector indicating the tape position $i'$, an indicator for $a' \in \{\text{WRITE}(\sigma) : \sigma\}$, and a scalar indicating the position $i'$.

$$\begin{pmatrix} e_{i'} \in \mathbb{R}^N \\ 1_{\exists \sigma : \delta_i = \text{WRITE}(\sigma)} \in \mathbb{R} \\ i \in \mathbb{R} \end{pmatrix} \tag{18}$$

The query, given input token $\xi = (i, a, q)$, is

$$\begin{pmatrix} 3N e_i \in \mathbb{R}^N \\ 2N \in \mathbb{R} \\ 1 \in \mathbb{R} \end{pmatrix} \tag{19}$$

If the tape position has previously appeared, the attention score will be maximized by the most recent position at which a write operation occurred; otherwise, it will fall somewhere else. We define the value, given input token $\xi' = (i', a', q')$, as

$$\begin{pmatrix} e_{i'} \\ e_{a'} \end{pmatrix} \tag{20}$$

The MLP then checks if the value is in

$$\left\{ \begin{pmatrix} e_i \\ e_{\text{WRITE}(\sigma)} \end{pmatrix} : \sigma \in \Sigma \right\} \tag{21}$$

If not, the symbol at tape position $i$ must be what it was set to before the computation started. Else, the symbol at the position must be as given by the action $a' \in \mathcal{A}$ retrieved.

Simultaneously, a second head attends to position $i$ and check whether it is part of the original input (i.e., precedes the separator); if it is, that position provides the symbol; if it is not, the tape position has not yet been written to.

Based on the information gathered by these two heads, the MLP then computes the next action, and outputs the new tape position, the action, and the new state.

**Comparison to other constructions**  Our construction is closest to that of Wei et al. (2022a). It is also similar to that of Qiu et al. (2024). The original Turing completeness proof for scratchpads (Pérez et al., 2019) recomputed the current tape position in every step, an idea used by Merrill & Sabharwal (2024a); our translation differs by keeping the tape position explicitly in the CoT, rather than recomputing it in every step; we believe that this reduces some technical challenges in the construction (at the price of making the transformer's width and parameters dependent on the input length). Bhattamishra et al. (2020b) noted that the classical RNN construction of Siegelman & Sontag (1995) can be straightforwardly replicated in a transformer using hard attention and unbounded-precision activations. This construction also is expressible in the UHAT model. A related translation, albeit in the special case where the Turing machine produces no repeated strings, is developed in Theorem 4.1 of Hou et al. (2024); hence our lower bounds also hold for the "Turing programs" scratchpad technique proposed by Hou et al. (2024). Another translation (applicable to, but not specific to transformers) is used by Malach (2024), it uses even simpler computations in each step, but may suffer a polynomial slowdown compared to Turing machines.

## C. Applications to Algorithmic Tasks

### C.1. PARITY and other Finite-State Languages

#### C.1.1. PROOF OF COROLLARY 4.3: CHARACTERIZATION FOR FINITE-STATE LANGUAGES

**Corollary C.1** (Restated from Corollary 4.3)**.** *Let $f$ be the membership problem of a finite-state language $L$. Then $f$ exactly one of the following holds:*

1. *$L \in AC^0$ and $f$ is expressible in UHAT without CoT*

2. *$L$ is not decidable in $AC^0$, and any UHAT CoT for $f$ has length $\Omega(N)$*

*Proof.* We show this using the characterization of regular languages in $AC^0$ by Barrington et al. (1992) and the follow-up result on hard-attention transformers by Yang et al. (2024).

First, if $L \in AC^0$, then it is definable in UHAT (without CoT) by Corollary 8 of Yang et al. (2024).

Conversely, if $L \notin AC^0$, then by Theorem 3 of Barrington et al. (1992), the syntactic morphism of $L$ is not quasi-aperiodic. That is, if $\eta_L$ is the syntactic morphism of $L^6$, then there is $t \in \mathbb{N}$ such that $\eta_L(\Sigma^t)$ contains a nontrivial group $G$. As in the proof of Theorem 3 of Barrington et al. (1992), we now consider an element $m \in G$ of order $k > 0$, and define $M' := \{m, m^2, \dots, m^k\}$ to be the subgroup defined by $m$; its identity element is $m^k$. We can then find strings $u, v \in \Sigma^t$ such that $\eta_L(u) = m, \eta_L(v) = m^k$.

Namely, there are strings $\alpha, \beta \in \Sigma^*$ such that either $\alpha u \beta \in L, \alpha v \beta \notin L$, or the other opposite holds. By passing from $L$ to its complement if needed, we may assume without loss of generality that $\alpha u \beta \in L, \alpha v \beta \notin L$. We now consider inputs of the form

$$\alpha(u|v)^* \beta \tag{22}$$

Now assume that a UHAT CoT with length $|g(x)| = o(|x|)$ exists for deciding, on such inputs, membership in $L$. By hard-coding the (fixed) strings $\alpha, \beta$, we can convert this into a UHAT CoT performing the same problem with only the inner part, a word in $(u|v)^*$, given as input. As $|u| = |v|$, we can further convert this into a UHAT CoT over a new alphabet $\Sigma' = \{a, b\}$, translating $u \in \Sigma^t$ to $a \in \Sigma'$ and $v \in \Sigma^t$ to $b \in \Sigma'$, cutting the string length by a constant factor of $t = |u| = |v|$. We now reach a contradiction: On inputs of the chosen form, checking membership in $L$ amounts to counting how often $u$ (or, after changing the alphabet, $a$) appears modulo $k$. Fixing any constant fraction of symbols cannot fix the result. By Theorem 3.3, the UHAT CoT cannot have had length $|g(x)| = o(|x|)$ in the first place. Hence, any UHAT CoT for deciding membership in $L$ has length $\Omega(N)$. $\square$

*Remark* C.2. We also mentioned, below Theorem 4.3, that $f$ is affected by the sensitivity-based barrier discussed in Section 2.1 if and only $L \notin AC^0$. This is seen as follows. First, if $L \in AC^0$, then by the result of Boppana (1997), $as_N(f) = \mathcal{O}(poly(\log(N))) = o(N)$. Second, if $L \notin AC^0$, then the deciding membership of strings from $\alpha(u|v)^* \beta$ as defined in the proof of Theorem 4.3 amounts to counting how often $u$ appears modulo $k$. Any function $\Sigma^* \to \{0, 1\}$ that exhibits this behavior on strings from $\alpha(u|v)^* \beta$ must have a high sensitivity. Overall, this means that shortcuts to finite-state automata without linear-length CoT (Liu et al., 2023) are likely to be learned well on long inputs if and only the automaton can be simulated in $AC^0$.

### C.1.2. EXPERIMENT

We trained a 2-layer 2-head transformer using the GPT-2 architecture (Radford et al., 2019b), with $d = 256$. Each model was trained at a fixed input and CoT length. Each training and testing input was generated randomly on the fly. Training was for 30K steps, at a batch size of 64, with AdamW (learning rate 1e-4).

### C.2. Arithmetic

#### C.2.1. PROOF OF THEOREM 4.5 (LOWER BOUND FOR MULTIPLICATION))

**Theorem C.3.** *A UHAT CoT for $M_N$ requires length $\Omega(N)$.*

*Proof.* In order to understand the difficulty of different digits in multiplication, we first review the connection between multiplication and bit string convolution. The integers $X, Y, XY$ can be written in terms of binary coefficients $\xi_i, \eta_i \in \{0, 1\}$ as:

$$X = \sum_{i=0}^{N} 2^i \xi_i$$

$$Y = \sum_{i=0}^{N} 2^i \eta_i$$

$$XY = \left(\sum_{i=0}^{N} 2^i \xi_i\right) \cdot \left(\sum_{i=0}^{N} 2^i \eta_i\right)$$

---

[6] We refer to Barrington et al. (1992) for the relevant definition.

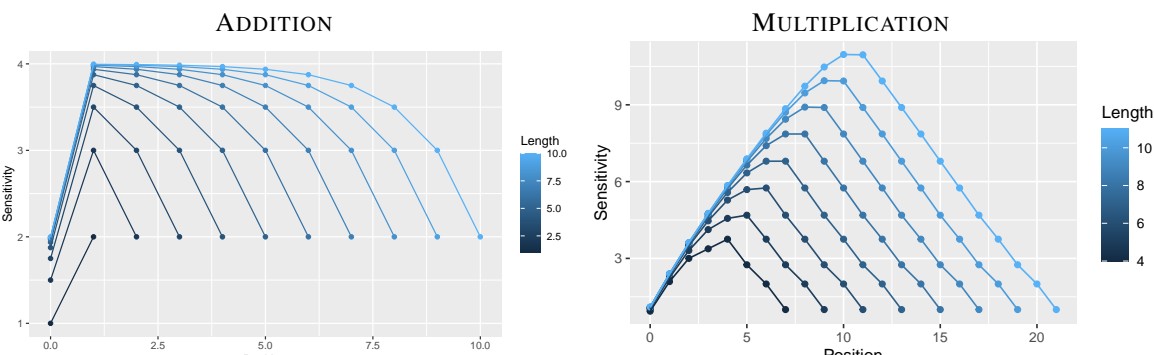

Figure 7: Arithmetic. Average sensitivity (Section 2.1) of the results digits as a function of the length $N$ of the operands (colored lines) and the position $k$ (x-axis). The most significant digit is at position $k = 0$.

Left: ADDITION. Sensitivity grows sublinearly with $n$.

Right: MULTIPLICATION. Sensitivity grows rapidly with the length of the multiplicands, and is highest in the middle, where sensitivity shows linear growth with $n$.

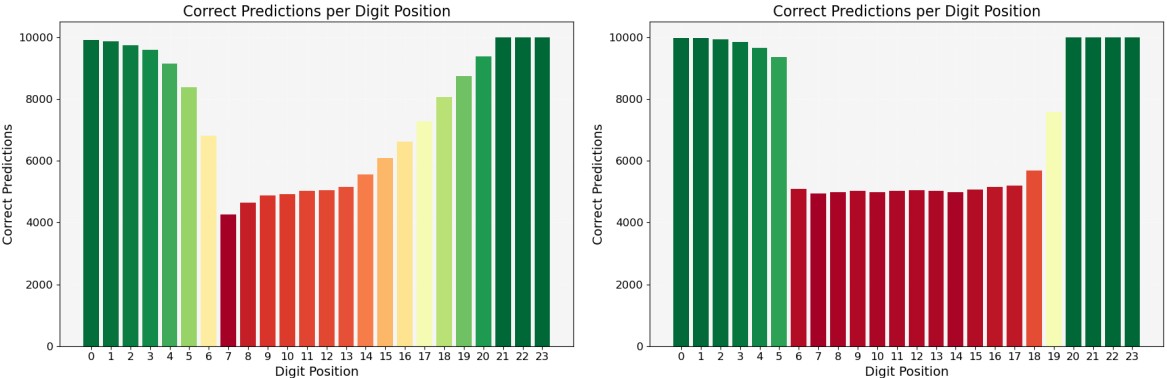

Figure 8: Accuracy of transformers trained on 12-digit multiplication (most significant digit at the left, zero-padded on the left), either autoregressively (left) or in parallel (right), on a test set with 10K number pairs. In both setups, the high-sensitivity digits in the middle (compare Figure 7 Right) show substantially decreased accuracy even if digits at beginning and end are predicted exactly.

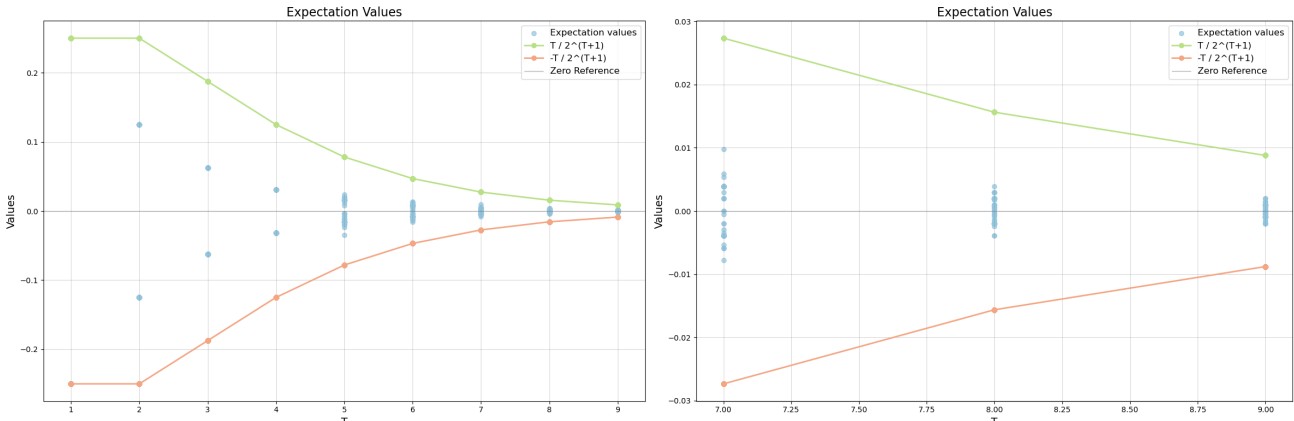

Figure 9: Numerical Experiments for Conjecture C.5: 100 different combinations samples for each T. Blue dots indicate values for (23) for 100 different sampled $(A, B, C)$. Right: Zoomed in on T = 7,8,9.

$$= \sum_{i,j} 2^{i+j} \xi_i \eta_j$$

$$= \sum_k 2^k \sum_{i+j=k} \xi_i \eta_j$$

Hence, the $N$-th digit of $XY$, $M_N$, is determined by, on the one hand, the high-order parity $\bigoplus_{i+j=k} \xi_i \eta_j$, XOR-red with carries obtained from terms at lower $k$'s. Now if one fixes $CN$ input bits, for $C$ small, the result cannot be fixed. $\qquad\square$

*Remark* C.4. Regarding $as(M_N)$, we note that the innermost digits $M_N$ are, up the impact of carries, derived from high-degree parities as discussed in the proof of Theorem 4.5. In accordance with this, we find that $as(M_k)$ is approximately $\min(k, 2N - k)$ and in particular peaks at $k \approx N$ (Figure 7). Empirically, the middle digits are the hardest for transformers (Figure 8).

### C.2.2. REMARK: AUTOREGRESSIVE MULTIPLICATION

Theorem 4.5 shows that querying individual digits of a product of $N$-digit numbers requires at least a linear-length CoT. An interesting question is whether autoregressive decoding can help, i.e., the earlier digits in the result can serve as an effective chain-of-thought for later digits. Empirical results cast doubt on such a possibility, because transformers struggle substantially more with multiplication than addition even in autoregressive generation (Figure 12).

Here, we outline a path to rigorously showing this. The key idea is the following conjecture, which is of purely combinatorial nature:

**Conjecture C.5.** *Given two binary encodings $X, Y \in \{0, 1\}^N$ of $N$-bit integers, let $M_i(X, Y)$ be the $i$-th bit of the product $XY$. We assume standard binary encoding with the most significant bit on the left.*

*Then, there is a constant $D > 0$ such that for any $T \in [1, N]$, sets $A, B \subseteq [1, N]$, $C \subseteq [1, T-1]$, we have*

$$\left| \mathbb{E} \left[ \hat{M}_T(X, Y) \prod_{i \in A} \hat{X}_i \prod_{j \in B} \hat{Y}_j \prod_{k \in C} \hat{M}_k(X, Y) \right] \right| \leq 2^{-D \cdot T} \tag{23}$$

*where $\hat{X}_i = 2X_i - 1 \in \{-1, 1\}$, and similar for $\hat{Y}$, $\hat{M}$.*

This statement intuitively states that digits of the product are not substantively correlated with combinations of individual digits in the inputs or preceding digits in the result. We provide numerical evidence for this conjecture in Figure 9. Any proof of this conjecture would imply lower bounds on autoregregressive multiplication. Formally, we are interested in autoregressive predictors

$$F_T : \{0, 1\}^{2N+T-1} \to \{0, 1\} \tag{24}$$

with the property that

$$M_T = F_T(X_1, X_2, \ldots, X_N, Y_1, Y_2, \ldots, Y_N, M_1, \ldots, M_{T-1}) \tag{25}$$

We then have bounds based on sensitivity (hence learnability by Section 2.1) and UHAT expressiveness:

**Lemma C.6.** *Assume Conjecture C.5. Then any Boolean function implementing $F_N$ needs to satisfy[7]*

$$as(F_N) = \Omega(N) \tag{27}$$

*A CoT directly outputting $M_1 \ldots M_{2N}$, without $\omega(1)$ further intermediate steps, cannot be implemented in UHAT.*

*Proof.* It is sufficient to show the lower bound on average sensitivity; the claim about UHAT follows from the inclusion of UHAT in $AC^0$ and Boppana's upper bound on the average sensitivity of $AC^0$ functions (Boppana, 1997). We refer to O'Donnell (2014) to background on Fourier analysis of Boolean functions. We consider the Fourier transform of $F_N$ as a Boolean function:

$$\hat{F}_N(X, Y, M_{1\ldots N-1}) = \sum_{A,B \subseteq [1,N]; C \subseteq [1,N-1]} \lambda_{A,B,C} \cdot \left[ \prod_{i \in A} \hat{X}_i \prod_{j \in B} Y_j \prod_{k \in C} \hat{M}_k \right] \tag{28}$$

for unique coefficients $\lambda_{A,B,C} \in \mathbb{R}$ given by

$$\lambda_{A,B,C} = \mathbb{E}_{X,Y} \left[ \hat{M}_N(X,Y) \prod_{i \in A} \hat{X}_i \prod_{j \in B} \hat{Y}_j \prod_{k \in C} \hat{M}_k(X,Y) \right] \tag{29}$$

We have, by the conjecture,

$$\lambda^2_{A,B,C} = \left( \mathbb{E} \left[ \hat{M}_N(X,Y) \prod_{i \in A} \hat{X}_i \prod_{j \in B} \hat{Y}_j \prod_{k \in C} \hat{M}_k(X,Y) \right] \right)^2 \leq 2^{-2DN}$$

Now, for any $\eta \in (0,1)$, the number of tuples $(A,B,C)$ where $|A| + |B| + |C| \leq \eta \cdot 3N$ is $\approx 2^{\gamma_\eta N}$ for some $\gamma_\eta > 0$. Hence,

$$\sum_{A,B,C:|A|+|B|+|C| \leq \eta \cdot 3N} |\lambda_{A,B,C}|^2 \lessapprox 2^{(\gamma_\eta - 2D)N}$$

By Parseval's Theorem:

$$1 = \sum_{A,B \subseteq [1,N]; C \subseteq [1,N-1]} \lambda^2_{A,B,C}$$

We choose $\eta$ so that $\gamma_\eta < 2D$. Then, by the link between Fourier transforms and average sensitivity (O'Donnell, 2014), we find:

$$
\begin{aligned}
as(F_N) &= \sum_{A,B,C} (|A| + |B| + |C|) \cdot |\lambda_{A,B,C}|^2 \\
&\geq 3\eta N \sum_{A,B,C:|A|+|B|+|C| \geq \eta \cdot 3N} |\lambda_{A,B,C}|^2 \\
&\geq 3\eta N \cdot (1 - 2^{(\gamma_\eta - 2D)N}) \\
&= \Omega(N)
\end{aligned}
$$

This concludes the proof. $\qquad \square$

---

[7]In fact, the stronger bound

$$as(F_T) = \Omega(T) \tag{26}$$

can be concluded uniformly for $T$, independently of $N$. This is because $F_T$ is only relatively weakly dependent on digits at positions much larger than $T$, making products trivially close to zero in Conjecture C.5 when $A$ or $B$ include indices substantially greater than $T$.

C.2.3. CONSTRUCTION OF $\mathcal{O}(N \log N)$-LENGTH COT FOR MULTIPLICATION

Whereas the traditional approach for multiplying two $N$-digit numbers requires $\Theta(N^2)$ steps, modern algorithms are more efficient for large inputs, and achieve an asymptotic runtime up to $\Theta(N \log N)$ (e.g. Schönhage, 1982; Harvey & Van Der Hoeven, 2021). We illustrate this by demonstrating a scratchpad whose length, $\mathcal{O}(N \log N)$, matches a widely conjectured lower bound on the asymptotic time complexity of multiplication in the Turing machine model. It is based on the Schönhage-Strassen algorithm (Schönhage, 1982).[8]

**Background: Number Theoretic Transform (NTT)**    The Number Theoretic Transform is an analogue of the Discrete Fourier Transform (DFT) that operates over finite fields. While the DFT leverages the complex roots of unity, the NTT employs roots of unity in a modular arithmetic setting. Formally, let $F_q$ denote a finite field of size $q$, where $q = p^k$ for some prime $p$ and integer $k$. Consider a prime $p$ such that there exists a primitive $n$-th root of unity $\omega \in F_q$, $\omega^n \equiv 1 \pmod{p}$ and for all $1 \leq k < n, \omega^k \neq 1$. The NTT is then defined as a mapping of an input vector of length $n$, say $a = (a_0, a_1, \ldots, a_{n-1})$, to an output vector $A = (A_0, A_1, \ldots, A_{n-1})$, where:

$$A_k = \sum_{j=0}^{n-1} a_j \omega^{jk} \pmod{p}, \quad \text{for } k = 0, 1, \ldots, n-1.$$

This transformation can be seen as a modular version of the classical DFT. The inverse of the NTT (INTT) is similarly defined, allowing for the reconstruction of the original vector from its transformed version. The INTT is expressed as:

$$a_j = \frac{1}{n} \sum_{k=0}^{n-1} A_k \omega^{-jk} \pmod{p}, \quad \text{for } j = 0, 1, \ldots, n-1,$$

where $\frac{1}{n}$ refers to the modular multiplicative inverse of $n$ modulo $p$, which must exist for the INTT to be valid. In the context of multiplication, the use of NTT allows for a scratchpad with $O(N \log N)$ length, compared to the $O(N^2)$ length of the naive scratchpad.

**CoT Construction and Analysis**    The Schönhage-Strassen algorithm provides a method to perform integer multiplication efficiently (Schönhage, 1982). To translate this into a scratchpad for use in a UHAT framework, the steps of the algorithm are defined here. We note that the required attention pattern is, for any input length $N$, input-independent, which ensures that the CoT is expressible in UHAT. The number of CoT steps scales as $\mathcal{O}(N \log N)$ because the NTT is computed using the divide-and-conquer Fast Fourier Transform strategy (Cooley-Tukey Algorithm, Cooley & Tukey, 1965).

We use "-1" as separator token.

**Input encoding**    We encode the input:

$$[-1] \; [\text{ First Number }] \; [-1] \; [\text{ Second Number }] \; [-1]$$

**Input Reversal and Padding**    Given two binary numbers, we first reverse their bit sequences and pad them to the nearest power of two, $2^n$, where $n$ is chosen such that $2^n$ is greater than or equal to the length of the input sequences. This ensures that the sequences are of a length suitable for efficient NTT computation.

$$[-1] \; [\text{ First Number Reversed and Padded }]$$
$$[-1] \; [\text{ Second Number Reversed and Padded }]$$

---

[8] The Schönhage-Strassen algorithm (Schönhage, 1982) requires $\mathcal{O}(N \log N \log \log N)$ steps, slightly different from the $\mathcal{O}(N \log N)$ CoT length. This is because we use an unbounded CoT alphabet to encode and perform arithmetic on the entries of the NTT. Moving to a constant-size CoT alphabet would incur a $\log \log N$ factor as in the standard Schönhage-Strassen algorithm. The algorithm of Harvey & Van Der Hoeven (2021) attains $\mathcal{O}(N \log N)$ complexity, but is substantially more complex.

**NTT Transformation** The padded sequences are then transformed using the NTT, which operates in a finite field. The output is a sequence $A_0, \ldots, A_{n-1} \in \{0, \ldots, p-1\}$.

$$[-1] \ [\, \text{NTT of First Number} \,] \ [-1] \ [\, \text{NTT of Second Number} \,]$$

This part takes $\mathcal{O}(N \log N)$ steps because the NTT is computed using the divide-and-conquer Fast Fourier Transform strategy (Cooley-Tukey Algorithm). Every step combines two fixed previously computed results; the attention pattern can be hard-coded into positional encodings and the modular arithmetic required can be hard-coded into the MLP map $f^{MLP}$ (Section 2.2).

**Convolution** In the NTT domain, the two transformed sequences are multiplied pointwise, modulo our prime number. This convolution corresponds to a multiplication in the original domain. We found that training was improved by first copying the NTTs with each digit marked with an index hint (Zhou et al., 2023) indicating its position, as a way of indicating which pairs of numbers to match in pointwise multiplication.

$$[-1] \ [\, \text{Convolution} \,] \ [-1]$$

This part takes just $\mathcal{O}(N)$ steps.

**Inverse NTT** After the convolution, we apply the inverse NTT to convert the sequence back to the domain.

$$[-1] \ [\, \text{Result of Inverse NTT} \,] \ [-1]$$

Again involving a divide-and-conquer FFT, this takes $\mathcal{O}(N \log N)$ steps.

**Recombination** Finally, we recombine the elements of the transformed sequence to produce the final multiplication result. Starting from the leftmost element, we multiply by $2^0$, the next by $2^1$, and so on, summing these weighted values to get the desired output.

$$[\, \text{Recombined Final Result} \,] \ [-1]$$

**Example** Below is an example of the full CoT for 2 digit binary multiplication (10 * 11) using prime $p = 5$.

- **Input**:

$$-1 \ \ 1 \ \ 0 \ \ -1 \ \ 1 \ \ 1 \ \ -1$$

- **Target**:

$$-1 \quad \underbrace{0 \ \ 1 \ \ 0 \ \ 0}_{\text{First Number Reverse and Pad}} \quad -1 \quad \underbrace{1 \ \ 1 \ \ 0 \ \ 0}_{\text{Second Number Reverse and Pad}} \quad -1 \quad \underbrace{1 \ \ 2 \ \ 4 \ \ 3}_{\text{NTT of First Number}}$$

$$-1 \quad \underbrace{2 \ \ 3 \ \ 0 \ \ 4}_{\text{NTT of Second Number}} \quad -1 \quad \underbrace{a \ \ 1 \ \ b \ \ 2 \ \ c \ \ 4 \ \ d \ \ 3}_{\text{NTT of First with Index Hints}} \quad -1 \quad \underbrace{a \ \ 2 \ \ b \ \ 3 \ \ c \ \ 0 \ \ d \ \ 4}_{\text{NTT of Second with Index Hints}}$$

$$-1 \quad \underbrace{a \ \ 2 \ \ b \ \ 1 \ \ c \ \ 0 \ \ d \ \ 2}_{\text{Convolution with Index Hints}} \quad -1 \quad \underbrace{2 \ \ 1 \ \ 0 \ \ 2}_{\text{Convolution}} \quad -1 \quad \underbrace{0 \ \ 1 \ \ 1 \ \ 0}_{\text{INTT of Convolution}} \quad -1 \quad \underbrace{0 \ \ 1 \ \ 1 \ \ 0}_{\text{Result (Recombined)}} \quad -1$$

## C.2.4. EXPERIMENTS

**Setup** All experiments were conducted on NVIDIA A100 GPUs (40GB memory each). We experimented with two different transformer architectures: a BART-based encoder-decoder model (for autoregressive decoding of multiplication) and an encoder-only model (for parallel decoding).

**Model Configurations**    We experiment with three setups:

1. Direct parallel decoding of each digit $M_1, \ldots, M_{2N}$ (and analogously for addition) in parallel. To avoid training separate transformers for each digit, we use a transformer encoder (i.e., transformer with bidirectional attention) reading in the two $N$-digit operands, and providing predictions for each of the $2N$ results digit at the top layer. Due to high sensitivity of the middle digits (Figure 7) and Theorem 4.5, we expect that this setup will be difficult for MULTIPLICATION, though not for ADDITION.

2. Autoregressive decoding of the result $M_1 \ldots M_{2N}$ (and analogously for addition). We expect that this setup is still difficult for MULTIPLICATION as discussed in Appendix C.2.2.

3. Autoregressive decoding of the $\mathcal{O}(N \log N)$-length CoT. We expect that this setup will make the task feasible for transformers.

*Encoder-only Model for Direct Parallel Decoding*: We employed a transformer encoder with 2 layers, 2 attention heads, a feed-forward network dimension of 512, and a model dimension of 128. The vocabulary size was determined by the tokenizer used for encoding the digits and arithmetic symbols.

*Encoder-Decoder Model for Autoregressive Decoding*: In autoregressive decoding, we slightly deviated from the theoretical setup in Section 2.2 in allowing bidirectional attention on the input (though not on the CoT) as in Abbé et al. (2024), for consistency with the Direct Parallel Decoding setup, where we use bidirectional attention for efficiency, as described above. We built on the BART architecture, with a 1-layer encoder and a 2-layer decoder. Both the encoder and decoder had 2 attention heads, and the decoder's feed-forward network dimension was 512. The model dimension for both encoder and decoder was 128.

*Note on Bidirectional Attention*: As the results in Hahn (2020) do not presuppose causal masking, our lower bounds are also applicable in the setup with bidirectiobnal attention (Remark B.7).

**Datasets**    All experiments were conducted on binary numbers. We trained and evaluated both models without CoT on datasets consisting of 2-digit, 4-digit, and 6-digit arithmetic problems for both addition and multiplication. For the CoT, we used a training dataset of 5 million samples and 10,000 test samples. The dataset sizes for 8-digit to 12-digit multiplication were smaller due to the limited number of possible combinations, with 50,000 samples for training and 10,000 for testing. In both cases, the test data is held-out, without overlap with the training data.

**Training Configurations**    For the CoT multiplication experiments, we used the following training configuration: - Number of training epochs: 20 - Batch size (per device): 8 for both training and evaluation - Gradient accumulation steps: 4 - Learning rate: 0.0001

| Multiplication Task | Training Set Size | Accuracy (%) |
|---|---|---|
| 8-digit | 50,000 | 99.83 |
| 9-digit | 50,000 | 99.87 |
| 10-digit | 50,000 | 99.82 |
| 11-digit | 50,000 | 99.85 |
| 12-digit | 5,000,000 | 99.84 |
| 13-digit | 5,000,000 | 99.82 |
| 14-digit | 5,000,000 | 99.96 |
| 15-digit | 5,000,000 | 99.91 |
| 16-digit | 5,000,000 | 99.83 |

Table 1: Accuracy results of the $\mathcal{O}(N \log N)$-length CoT for MULTIPLICATION for different digit sizes. All accuracies are computed on tests sets disjoint from the training set.

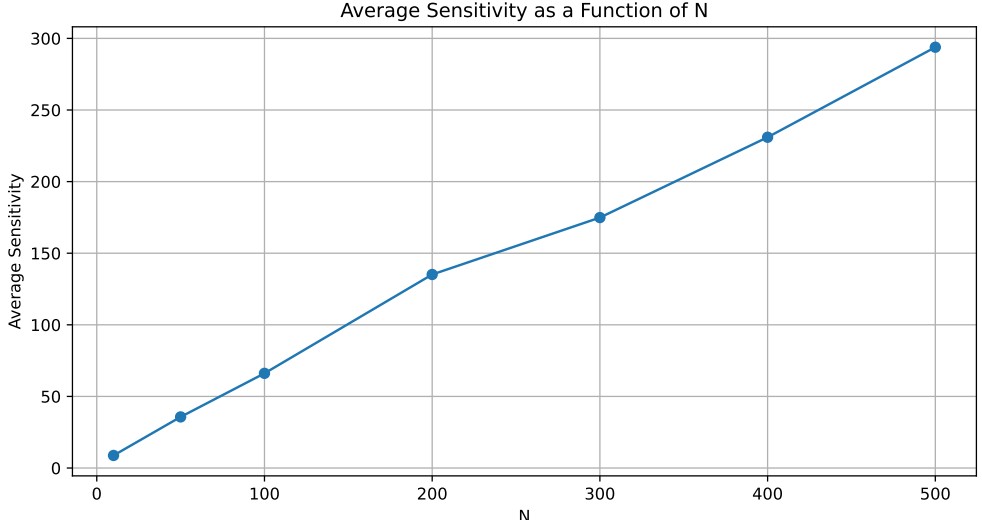

Figure 10: Average sensitivity of the last digit of MEDIAN as a function of $N$ when $B \sim 1 + \lceil \log N \rceil$. We estimate average sensitivity by sampling 200 input strings at each $N$, and 200 bit flips for each input. As predicted theoretically (Appendix C.3), average sensitivity grows linearly in this regime where $\exp(B) \gtrsim N$.

## C.3. MEDIAN Task

### C.3.1. THEORY

We first establish the claim that the last digit of MEDIAN is sensitive to changes of the integers, in the regime where $N << \exp(B)$.[9] Changing the first digit in any of the numbers with order rank $< \lfloor \frac{N}{2} \rfloor$ can get it to have rank $> \lfloor \frac{N}{2} \rfloor$. Under uniform sampling of the numbers, the chance that this will change the last digit is $\approx \frac{1}{2}$. Changing other digits will additionally have some nonzero (though smaller) chance of flipping the last output digit. Hence, the average sensitivity of the last digit is $\gtrsim \frac{N}{2}$. This is illustrated in Figure 10. Hence, the task is challenging for transformers by the results reviewed in Section 2.1.

Now we proceed to showing the CoT bounds.

**Theorem C.7** (Restated from Theorem 4.8). *For the MEDIAN task, a UHAT scratchpad requires length $\Omega(N)$. This bound is attained.*

*Proof.* We note that the input length is $BN$, where $B$ is the number of bits in each integer. To show the lower bound by applying Theorem 3.3, we note that fixing, say, $\frac{1}{10}N = \frac{BN}{10B}$ digits cannot fix the median.

Now we show that the bound is attained. Recall that the input consists of $N$ unique numbers in binary encoding, each with $B$ bits. Let $a_1, \ldots, a_N$ be the integers in the input, and let $a_{i_1}, \ldots, a_{i_N}$ be the same numbers ordered by magnitude. Now the CoT simply consists of

$$a_{i_1} \ldots a_{i_{\lfloor N/2 \rfloor}} \tag{30}$$

To implement this in UHAT, we first use $B$ attention heads in order to aggregate each $a_i$ on the basis of the $B$ bits entering its binary representations:

$$a_i = \sum_{j=0}^{B-1} 2^j \lambda_{B(i-1)+j+1} \tag{31}$$

where $\lambda \in \{0,1\}^{BN}$ is the input string consisting of $N$ integers with $B$ digits each. This number $a_i$ is stored, e.g. as a one-hot encoding, in the activation $\boldsymbol{y}^{(1)}$ at the last digit of its binary representation in the input. The CoT then orders the

---

[9]If $N$ is on the order of or even larger than $\exp(B)$, then uniform sampling of numbers (even with replacement) will have a high chance of producing the same number and low sensitivity. We hence focus on the regime where $N << \exp(B)$.

numbers by magnitude, stopping at the $k$-th index. Importantly, this is expressible with a single attention head as follows: If the integer $i$ is represented as the one-hot vector $e_i$, then $e_i^T K^T Q e_j$ produces the desired behavior when[10]

$$(K^T Q)_{ij} = \begin{cases} -1 & \text{if } i \leq j \\ N + j - i & \text{else} \end{cases} \tag{32}$$

Stopping at $\lfloor N/2 \rfloor$ can be hard-coded via positional encodings. □

### C.3.2. EXPERIMENT

To simplify the setup, we deviated from the theoretical construction by using decimal numbers instead of binary numbers, and also using decimal numbers instead of atomic encodings in the CoT. We considered three-digit numbers ($B = 3$, but with decimal instead of binary encoding).[11]

We experiment with $N = \{1, 9, 19, 29, 39, 49, 59, 69, 79, 89, 99\}$. The resulting input length is between 6 and 398 (including BOS and SEP token). The $N$ numbers are randomly sampled from $\{0, 1, \cdots, 999\}$ with replacement. We train separate models for each $N$, and test on the same length. Note that the size of the test set varies with $N$, concretely it is $(N + 10) \times 20$. Because when $N = 1$, there are only 1000 possible inputs. The test examples are excluded from training set. The input format is as follows (for the case $N = 3$), including full scratchpad:

| BOS | 3 | 4 | 3 | ; | 0 | 1 | 9 | ; | 8 | 5 | 2 | ; | SEP | 0 | 1 | 9 | ; | 3 | 4 | 3 | ; | EOS | .

We use ";" to separate each number. As mentioned, we experiment with the full scratchpad and without any scratchpad, as well as every $k$th element in the full scratchpad, where $k = \{2, 3, 4, 5, 6, 9, 12\}$. The models are trained with cross entropy loss on tokens after SEP (excluding itself). The input length in figure 5 refers to number of tokens before and including SEP, and CoT length refers to number of token after SEP and before the final answer (last 3 digits). While models are trained to predict both the scratchpad and the answer, we do not consider scratchpad during testing. In other words, to evaluate the models, we give them tokens up to SEP, and let them generate with greedy search, and take the last generated number before EOS as the prediction. The prediction is considered correct only when it matches all digits in the answer. We run each experiment (each $N$ and each kind of scratchpad) with 3 random seeds, and report the average accuracy.

Regarding hyperparameters, we use the same model architecture for all experiments in this section, which has 3 layers, 4 attention heads in each layer, and model dimensionality of 256. We use batch size of 64, train models for 50k steps, use learning rate starting from $3 \times 10^{-4}$ and decreasing linearly. We use AdamW optimizer, with $\beta_1 = 0.9, \beta_2 = 0.999$ and weight decay of 0.01. All dropout rates are set to zero. Note that we observe that low accuracy is always accompanied with big training loss.

### C.4. Graph Reachability

### C.4.1. PROOF OF THEOREM 4.10

**Theorem C.8** (Restated from Theorem 4.10). *There is a family $\mathfrak{G}$ of DAGs inside which reachability is solvable in $TC^0$, but cannot be represented by a transformer at sublinear average sensitivity. A UHAT CoT needs length $\Omega\left(|E| \log |V|\right)$. This bound is attained.*

*Proof.* For the lower bound, the proof proceeds by coding $\text{PARITY}_n$ into DAGs with $2n$ vertices. For $i = 1, \ldots, N+1$, we introduce two vertices $v_{i,odd}$ and $v_{i,even}$. Whenever $x_i = 1$, we add edges $v_{i,odd} \to v_{i+1,even}$ and $v_{i,odd} \to v_{i+1,even}$. When $x_i = 0$, the edges instead connect $v_{i,odd} \to v_{i+1,odd}$ and $v_{i,even} \to v_{i+1,even}$. Then, $x$ has even parity if and only if there is a path from $v_{1,even}$ to $v_{N+1,even}$. We define $\mathfrak{G}$ as the set of these graphs. If $G$ is guaranteed to be in $\mathfrak{G}$, then a $TC^0$ circuit is sufficient for deciding membership. As the graphs in $\mathfrak{G}$ code $\text{PARITY}$, reachability cannot be represented by transformers at low sensitivity. Also, fixing a small constant fraction of edges, while staying within $\mathfrak{G}$, cannot fix the reachability.

---

[10]We note that definitions of autoregressive sorting by transformers are also given in RASP-L (Zhou et al., 2023) and C-RASP (Huang et al., 2024).

[11]In preliminary experiments, we found that it is important to have diverse inputs (i.e., $B$ not too small in relation to $N$) to avoid learning a "increment-by-1" algorithm (imagine when $N = 100$, we sample 100 number from [0, 99] without replacement, which means the sorted list and the median is always the same).

To show that the $\Omega\left(|E|\log|V|\right)$ bound is attainable, we note that a CoT of length $\mathcal{O}(|E|\log|V|)$ can encode breadth-first search, a generalization of the DFS/BFS CoT for the cycle task in Abbé et al. (2024). The input consists of the edges and the query, such as:

```
BOS 06 08 ; 04 01 ; 04 05 ; 01 03 ; 08 00 ; 01 05 ; ... QUERY1 03 QUERY2 04 SEP
```

Whereas the input codes each vertex as a $\lceil\log|V|\rceil$-length number (keeping the input alphabet finite), each vertex is coded as an atomic token inside the CoT. At the beginning of the CoT, each edge is translated from binary representations to this atomic representation. This takes $\mathcal{O}(|E|\log|V|)$ steps.[12]

We then implement a standard first-in-first-out queue. As nothing can be deleted from the CoT sequence once it has been generated, we maintain a pointer $\in \mathbb{N}$ that indicates the current head of the queue. The CoT then starts with the first query vertex. We copy into the CoT all the edges starting with the end of the edge. We then move to the second edge in the CoT and write down all the edges starting with the end of that edge, and so on. If at any point we encounter the target vertex, we exit. Throughout, in every CoT step, we additionally maintain the pointer $\in \mathbb{N}$ indicating the current head of the queue. □

### C.4.2. EXPERIMENT

We show that the CoT described in Theorem C.8 is sufficient for Transformers to solve the problem of DAG reachability in the general case. For that, we train Transformers to predict the reachability of two vertices in a random DAG, both with and without a CoT.

**Data generation.** Each random DAG is generated by sampling a random lower triangular adjacency matrix and instantiating a DAG from it. We then compute the distances between all pairs of vertices in the graph. When sampling examples for training or evaluation, we select, with equal probability, either an unconnected pair labeled as 0 or a connected pair labeled as 1. For connected pairs, we sample the pairs such that the distribution of possible distances is uniform.

For example, in a chain graph $A \to B \to C$, the unconnected pairs $(C, B)$, $(C, A)$, and $(B, A)$ are sampled with a probability of 1/6 each. The connected pairs $(A, B)$ and $(B, C)$ are sampled with a probability of 1/8 each (distance = 1), while the pair $(A, C)$ is sampled with a probability of 1/4 (distance = 2). This way, we avoid the bias toward shorter distances.

For each generated DAG, we compute the Weisfeiler-Lehman hash and ensure that the hashes of the training and test DAGs do not overlap. When provided to the model, the input data is encoded as described in Theorem C.8.

**Model and training.** We use a decoder-only Transformer based on the GPT-2 architecture (Radford et al., 2019a). The model has 4 layers, 4 attention heads per layer, and 256 hidden dimensions. The model is trained for 50k steps using AdamW with a batch size of 64 samples.

We train the models in two regimes: with and without a CoT. In the CoT regime, we generate a BFS-based CoT as described in Theorem C.8 and append it to the input data. To simplify and reduce computational cost, we represented all vertex indices as decimal numbers, eschewing conversion to atomic symbols. During training, we optimize the next-token prediction loss on the CoT and answer parts of the sequence, while ignoring the predictions for the input tokens. During evaluation, the model autoregressively generates the CoT and the final prediction. The no-CoT regime is similar, but the CoT is not appended to the input, requiring the model to generate the answer directly.

Evaluation accuracy is checked every 9k steps. If the evaluation accuracy exceeds 99.5%, training is terminated early.

**Results.** The model's accuracy for various input sizes, corresponding to DAGs with 5 to 35 vertices, is shown in Figure 4. All values are averaged over three runs.

For all graph sizes except one, Transformers with CoT achieve near-perfect accuracy, while Transformers without CoT perform at chance level. The exception is for graphs with the smallest size (5 vertices), where both regimes achieve approximately 90% accuracy. This discrepancy may be due to the limited number of distinct DAGs of this size, leading to insufficient training signal for the model to learn the algorithm of CoT construction.

---

[12]Formally, $a = \sum_{i=0}^{B}\lambda_i 2^i$, define $\widehat{a}_j := \sum_{i=0}^{j}\lambda_i 2^i$. The CoT then translates the binary representation of each vertex index $a \in [1, |V|]$ into the atomic representation by enumerating $0\widehat{a}_{1\,0} \ldots \widehat{a}_{1\,\lceil\log|V|\rceil}$

# D. Experiments with Pretrained LLMs

**Approach.**   To test our predictions on the necessary CoT length, we run experiments with state-of-the-art LLMs trained to generate CoT reasoning before responding to a user's request: DeepSeek-R1 (Guo et al., 2025) and o1-mini (Jaech et al., 2024). If, contrary to our predictions, a sub-linear algorithm for any of the discussed problems exists, these models might discover it and solve the task with a CoT of sub-linear size. Verifying this serves as a basic sanity check for our theory.

We tested the models on three tasks: parity, multiplication, and median. For the parity task, we generated random bitstrings of various lengths, ranging from 10 to 70, and asked the models to calculate their parity. We then selected the CoTs that led to correct answers and calculated their average size for each input length. The prompt provided to both models was: `You will receive a string.  You have to manually calculate its parity. Finish your response with 1 if the parity is odd, and 0 if the parity is even.`

The approach for the multiplication task was similar. We generated random pairs of numbers with lengths ranging from 3 to 9 digits, prompted the models to multiply them, and computed the sizes of the correct CoTs. The prompt provided was: `You will receive two numbers.  You have to multiply them manually. Finish your response with the precise result of the multiplication.`

For the median task, we generated sequences of odd length and prompted the models to find the median. The numbers were generated uniformly from 1 to $10^5$. The prompt provided was: `You will receive a sequence of numbers. You have to manually compute its median.  Finish your response with the value of the median of this sequence.`

**Results.**   The results of the experiment are shown in Figure 6. For all three tasks, the size of the CoTs grew at least linearly, supporting the theoretical prediction that no CoT exists to solve parity, multiplication, or median in a sub-linear number of steps.

A qualitative inspection of the reasoning traces of DeepSeek-R1 revealed that, when computing the parity of a bit sequence, it copied every bit, counted the number of ones, and then determined the evenness of that count. For multiplication, it used a naive quadratic algorithm. However, the performance of DeepSeek-R1 dropped significantly for numbers with 10 digits or more, suggesting that its multiplication algorithm is not truly length-generalizable.

# E. Further Discussion on Related Work

## E.1. CoT Constructions in the Literature

Besides CoTs emulating Turing machines (Appendix B.3), various other more specialized CoT constructions have been considered in the literature. Feng et al. (2023) provide a general construction of transformers for a very broad class of dynamic programming (DP) algorithms. This construction essentially only uses hard-attention operations, and can be expressed in UHAT as long as the aggregation function used in the DP algiorithm can be expressed. This includes, for instance, evaluating Boolean formulas. Cabannes et al. (2024) analyzed the algorithm learned by a transformer trained on scratchpads for various algorithmic problems, identifying an "iteration head" attention pattern. This head effectively computes a unique hard attention pattern iterating through the input, combining material from the input with the last CoT step. Relatedly, the Inductive Scratchpad Abbé et al. (2024) is a construction where each step is a function specifically of the previous CoT step and the input.

## E.2. Relation to Globality Degree

Here, we discuss the relation of our results to the work by Abbé et al. (2024) on the Globality Degree. We recall their definition, rephrased for self-containedness:

**Definition E.1** (rephrased from Definition 2 of Abbé et al. (2024)).   For an alphabet $\mathcal{A}$ of cardinality $\mathcal{O}(poly(n))$, and a distribution $D$ on $\mathcal{A}^n \times \mathcal{A}$, $D$ has *constant globality* if and only if there is $k \in \mathbb{N}$ such that, for each $n$, there exists $S \subseteq [n]$, $|S| = k$, such that

$$\mathrm{I}\left[(X[S], \hat{P}_X); Y\right] = n^{-O(1)} \tag{33}$$

where $(X, Y) \sim D$, and $\hat{P}_X$ is the histogram of tokens in $X$.

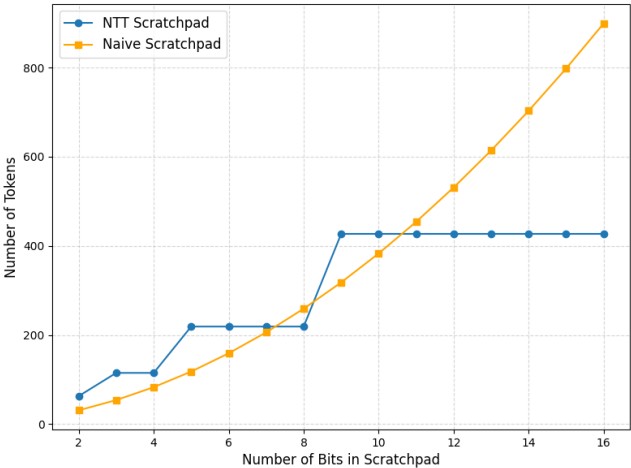

Figure 11: Multiplication: Comparing the number of tokens used in the $\mathcal{O}(N \log N)$ NTT CoT (Appendix C.2.3) vs a CoT implementation of the naive $\mathcal{O}(N^2)$ algorithm. The token count of the NTT CoT is piecewise constant because the length only depends on the prime $p$ used in the construction; length is asymptotically $\mathcal{O}(N \log N)$.

Here, we think of $(X, Y) \in \mathcal{A}^n \times \mathcal{A}$ as a pair of input and label; hence, $Y$ will generally be a function of $X$.

If one assumes $|\mathcal{A}| = O(1)$ and puts aside the knowledge of the token histogram, then constant globality is equivalent to the presence of nontrivial correlations with the circuit complexity class $NC^0$ (Lemma 6 in Abbé et al. (2024)), a highly restricted class contained within $AC^0$ and easily simulated by UHAT.

The key conjecture of Abbé et al. (2024) states:

**Conjecture E.2** (Conjecture 1 of Abbé et al. (2024)). *A distribution $P_{X,Y}$ with well-behaved $P_X$[13] is efficiently weakly learnable by a T-regular Transformer[14] if and only if $P_{X,Y}$ has constant globality.*

For CoTs, the implied prediction is that (i) tasks with nonconstant globality require a CoT for efficient learning, (ii) a CoT is learnable if and only if each of its intermediate steps has constant globality. A version of Conjecture E.2 is shown formally for a cycle classification task, though the conjecture makes far-reaching predictions beyond that. The practical predictions made by this conjecture are related to those of our theory, but not equivalent; for instance:

1. (diverging prediction) PARITY has low globality, because the conditioning on the histogram is not constrained. Conjecture E.2 thus provides a more optimistic prediction here than our results.

2. (converging prediction) The Cycle task of Abbé et al. (2024) requires a linear-length CoT by Theorem 3.3, and can also be shown to not be representable at sub-linear average sensitivity under a reasonable input encoding. Difficulty is predicted under both perspectives.

Overall, there are similarities and differences between the practical predictions made by Conjecture E.2 and our unconditional lower bounds. On a technical level, the theoretical arguments of Abbé et al. (2024) rely on *bidirectional* attention at least on the original input (if not the scratchpad), different from the causally masked transformers generally used in LMs. Expanding those arguments to causally masked transformers is an interesting problem for further research.

---

[13]This is defined in Definition 6 in (Abbé et al., 2024). A simple example is the uniform distribution over $\{0, 1\}^N$.

[14]Informally, this is a bidirectional transformer under a standard initialization (Definition 4 in (Abbé et al., 2024)).

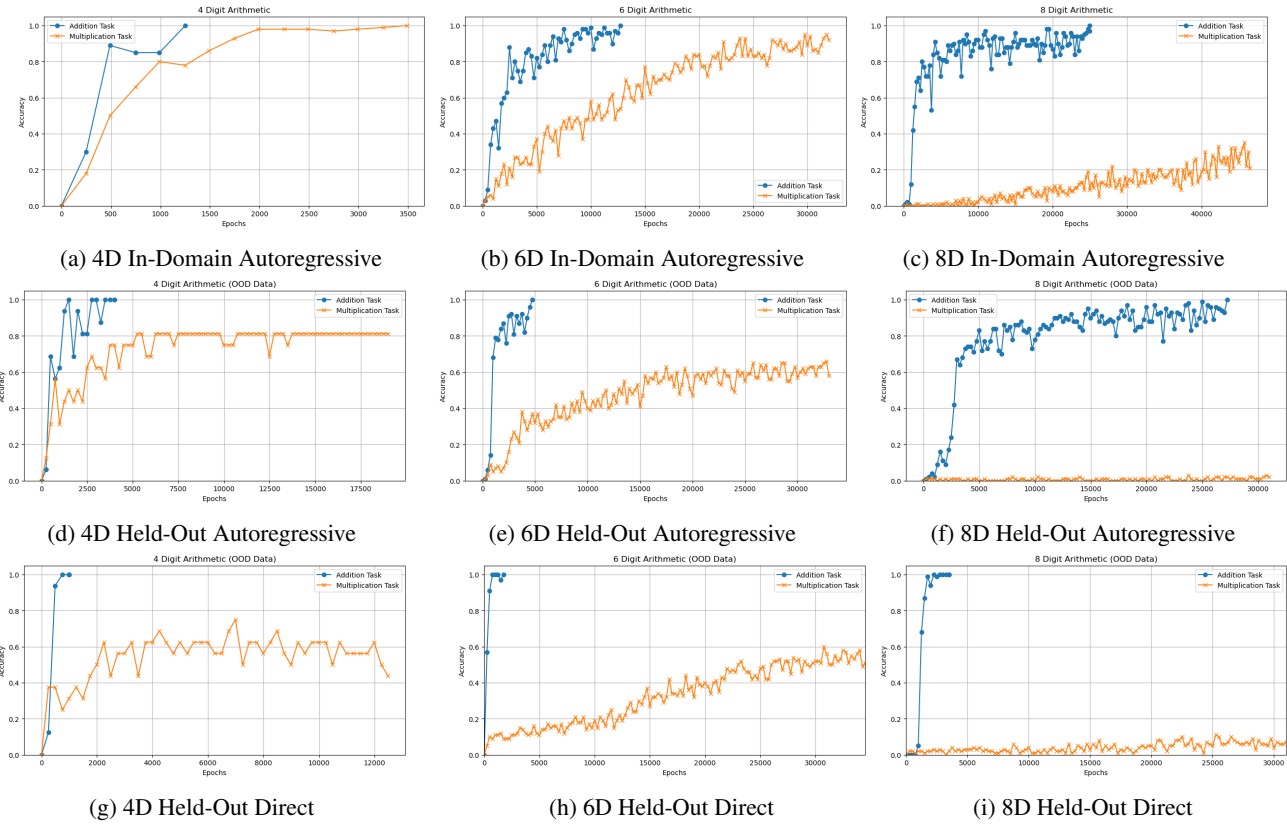

Figure 12: Arithmetic: Results on ADDITION and MULTIPLICATION of binary numbers, no CoT. Top row: In-Domain test data, Middle row: Held-Out test data (autoregressive decoding), bottom row: Held-Out test data (direct parallel decoding). Addition is learned well, even when the digits are directly decoded. In contrast, multiplication does poorly.

