# OpenReview forum: "Lower Bounds for Chain-of-Thought Reasoning in Hard-Attention Transformers"
_ICML.cc/2025/Conference — ICML 2025 poster_

### Official Review · Reviewer_M2re · 2025-03-12

**Overall Recommendation:** 3

**Summary:**

This paper explores the use of CoT reasoning and scratchpads in enhancing the computational capabilities of transformers. The authors propose new lower bounds for the number of CoT steps required for various algorithmic problems, challenging optimistic bounds from circuit complexity.

**Claims And Evidence:**

The claims are supported by clear and convincing evidence.

**Essential References Not Discussed:**

In fact, there has already been some detailed discussion on CoT Boundaries in previous work. Please analyze in detail the differences between the following works and your proposed boundary:
1. Towards revealing the mystery behind chain of thought: A theoretical perspective. NeurIPS 2023
2. Unlocking the Capabilities of Thought: A Reasoning Boundary Framework to Quantify and Optimize Chain-of-Thought. NeurIPS 2024

Furthermore, there is a lack of significant work focusing on the interpretability of CoT. Relevant works include:
1. Wang et al. How Large Language Models Implement Chain-of-Thought? Arxiv 2023.
2. Hanna et al. How does gpt-2 compute greater-than?: Interpreting mathematical abilities in a pre-trained language model. NeurIPS 2023.
3. Dutta et al. How to think step-by-step: A mechanistic understanding of chain-of-thought reasoning. TMLR 2024.

**Experimental Designs Or Analyses:**

The conclusion of this experiment does not seem to indicate that O(n) is the lower bound of the complexity. At the same time, the conclusions of these experiments appear to be similar to those in [1, 2].

[1] Towards revealing the mystery behind chain of thought: A theoretical perspective. NeurIPS 2023

[2] Unlocking the Capabilities of Thought: A Reasoning Boundary Framework to Quantify and Optimize Chain-of-Thought. NeurIPS 2024

**Methods And Evaluation Criteria:**

I believe the analysis method is overly restrictive, particularly with the use of hard transformers and 0-1 digit operations. It seems difficult to accurately represent real-world scenarios using these approaches.

Furthermore, I think the model completely neglects the logical progression inherent in CoT reasoning. Treating all outputs as equivalent tokens seems problematic, especially for long dialogues where the summary might be treated in the same way. This approach doesn't appear to be reasonable.

**Other Comments Or Suggestions:**

None.

**Other Strengths And Weaknesses:**

None.

**Questions For Authors:**

- Could you please elaborate on the specific differences between this work and previous research mentioned above?
-  The experimental results do not seem to conclusively demonstrate that O(n) is the lower bound of the complexity. It would be helpful to see a more rigorous argument or further evidence supporting this claim.
- The assumptions made in this work appear to be quite strong, potentially even stronger than those in the original work by Feng et al. [1]. It might be useful to provide additional justification or relax some of these assumptions, to ensure the findings are robust and applicable in a wider range of scenarios.

[1] Feng et al. Towards revealing the mystery behind chain of thought: A theoretical perspective. NeurIPS 2023

**Relation To Broader Scientific Literature:**

None.

**Theoretical Claims:**

I have examined the proof process. The proof contains numerous assumptions that are either arbitrary or difficult to relate to practical contexts. For example, the assumption |{i ≤ N : ρN (i) = ∗}| ≥ CN is completely unclear in terms of its rationale, and the corresponding proof seems to fail to justify this core inequality.

Additionally, there are many unexplained symbols, such as *, ρ|x|, and |{i ≤ N : ρN (i) = ∗}|.

---

> ### Author Rebuttal · Authors · 2025-03-31
>
> We thank the reviewer for acknowledging that our claims are supported by clear and convincing evidence. We now address all the concerns mentioned in the review:
>
> 1. On the restrictiveness of our approach due to the use of hard attention and binary input/output format:
>
>     We agree that hard attention is a simplifying assumption, though it is amply supported by interpretability studies as we cited in Appendix E.1.
>
>     Perhaps even more importantly, recent results also show that hard attention has the *same expressiveness* as the real-world transformer setting -- that is, UHAT can simulate the output of finite-precision soft attention transformers with causal masking [3]. Hence, real-world transformers cannot have asymptotically shorter CoTs than UHAT transformers. Thus, our results *provably transfer to the real-world setup*. We will highlight this in the final version.
>
>
>     Regarding 0-1 digit operations, this is not a substantive restriction. The random restrictions technique has typically been applied to binary input strings in the past, which we follow for simplificity. However, the technique equivalently applies to strings over other alphabets, and, trivially, any finite alphabet can be encoded in binary. We will add explicit discussion, and a formal statement of the equivalence, in the final version of our paper.
>
> 2. On neglecting the logical progression in CoT reasoning:
>
>     Our bounds hold irrespectively of the contents of the CoTs, including their logical structure. If the concern of the reviewer is that some of the tokens in the CoT may be less important than others, then it is not a problem for our proofs, since we treat all tokens as important.
>
> 3. On arbitrary assumptions, specifically $|{i ≤ N : ρ_N (i) = ∗}| ≥ CN$:
>
>    We would like to point out that this is not an assumption; this is a property of a specific class of functions in Theorem 3.3. It formalizes the idea that a function stays constant on a set of strings "ignoring" many positions in those strings. Informally, if $|{i ≤ N : ρ_N (i) = ∗}| ≥ CN$ then with probability at least $C$ changing one bit in the input of $f$ won't affect the output; in other words, $f$ is insensitive. We will expand our informal explanation of the rationale behind the statement of the theorem. We will also be happy to add intuition on the meaning behind other assumptions if the reviewer clarifies which of them seem arbitrary.
>
> 4. On unexplained symbols:
>
>     Note that the * in Definition 3.2 is just a plain symbol in the output space of restrictions: $\rho_N : [1,N] \rightarrow \Sigma \cup \{*\}$, also used in the condition $\rho_{|x|}(i) \neq *$.
>     Other symbols are also either defined or standard notation, but we will take care to make the notation more accessible for a broader audience in the final version.
>
> 5. On experiments not supporting the statement that $O(n)$ is the lower bound of the complexity:
>
>     We assume that by the sentence "O(n) is the lower bound of the complexity" the reviewer references the group of our statements imposing the $\Omega(N)$ bounds on the length of CoT required to solve specific tasks. Generally, our experiments supporting this group of statements show that all successful CoT strategies for the tasks we consider require at least $\Theta(N)$ steps. We agree that it does not prove that no other successful sublinear CoT strategy exists; however, proving such statement experimentally is challenging since it isn't feasible to test all possible CoTs. We are happy to extend our experimental design if the reviewer has any suggestions.
>
> 6. On relation to [1] (Feng et al. 2023) and [2] (Chen et al. 2024):
>
>     The key difference to prior work on the theory of CoT is that our results provide provable lower bounds on the lengths of CoTs. This is the key advance compared to [1], which instead focused on showing that certain tasks require CoTs, but didn't study how many steps are minimally needed.
>
>     [2] shows the relationship between the type of the CoT and the maximum difficulty of the task that an LLM can solve with that CoT. While these results are important, they leave the concept of difficulty undefined, and they also do not provide the bounds on the required length of the CoT. Therefore, while the results of [2] are generally applicable to more tasks, our work offers much more precise and rigorous results for specific tasks.
>
> 7. On discussing interpretability of the CoT:
>
>     In the camera-ready version of the paper, we will include a discussion of the work on interpretability of CoT and cite the papers mentioned by the reviewer.
>
> 8. On excessively strong assumptions:
>
>     We ask the reviewer to clarify which assumptions are referenced in this concern and which of them are stronger than those in [1]. It is hard for us to give an answer to this question without knowing details.
>
> ---
>
> [3] Jerad et al. "Unique Hard Attention: A Tale of Two Sides." arXiv preprint arXiv:2503.14615 (2025).

---

### Official Review · Reviewer_PLKX · 2025-03-12

**Overall Recommendation:** 4

**Summary:**

This paper establishes lower bounds on the required chain-of-thought (CoT) length that unique hard-attention (UHAT) transformers need for solving certain classes of problems.
In particular, lower bounds are established for PARITY ($\Omega(N)$), MULTIPLICATION ($\Omega(N)$), MEDIAN ($\Omega(N)$), and REACHABILITY ($\Omega(\lvert E \rvert \log \lvert V \rvert)$), where $N$ is the length of the input.
Empirical experiments on both synthetic setups and LLMs validate the paper's claims.

**Claims And Evidence:**

Yes

**Essential References Not Discussed:**

None that I'm aware of.

**Experimental Designs Or Analyses:**

Yes

**Methods And Evaluation Criteria:**

Yes

**Other Comments Or Suggestions:**

N/A

**Other Strengths And Weaknesses:**

Strengths:
* This paper is quite solid and improves our understanding of transformer expressivity.
* Complexity lower bounds are appreciated, as they tend to be harder to prove than upper bounds.
* The list of FAQs in Appendix A resolves many of the questions I initially had.
* This is a good paper.


Weaknesses:
* I do not see clear weaknesses in this paper.

**Questions For Authors:**

N/A

**Relation To Broader Scientific Literature:**

This improves our understanding of LLM expressivity.

**Theoretical Claims:**

Yes

---

> ### Author Rebuttal · Authors · 2025-03-31
>
> We thank the reviewer for the positive assessment of our paper. We particularly appreciate the reviewer's point that complexity lower bounds, as shown in this paper, tend to be harder to prove than upper bounds.
>
> If possible, we'd like to kindly ask the reviewer to provide more justification supporting their score. We are concerned that, as the review is rather brief, it might be disregarded by the AC and other reviewers.

---

### Official Review · Reviewer_Does · 2025-03-16

**Overall Recommendation:** 2

**Summary:**

This paper establishes that hard-attention transformers require chain-of-thought (CoT) sequences of length linear in the input size to solve high-sensitivity algorithmic tasks like MEDIAN, and REACHABILITY in layered DAGs, with bounds tight up to logarithmic factors. By leveraging sensitivity analysis and a novel application of random restrictions, the authors prove that sublinear CoT steps would render these functions reducible to constants, which is impossible due to their inherent sensitivity, and further show that "dot-by-dot" CoTs require super-polynomial lengths. Empirical validation on synthetic tasks confirms sharp accuracy declines with sublinear CoT, while tests on real LLMs corroborate the necessity of sufficient intermediate steps.

**Claims And Evidence:**

All claims made in the submission are supported by clear and convincing evidence.

**Essential References Not Discussed:**

N/A

**Experimental Designs Or Analyses:**

The experimental designs are valid.

**Methods And Evaluation Criteria:**

The experiments make sense.

**Other Comments Or Suggestions:**

The template seems to be wrong.

**Other Strengths And Weaknesses:**

**Strengths**
1. **Theoretical Rigor**: The paper provides a foundational analysis using tools from circuit complexity (e.g., sensitivity, random restrictions) to derive lower bounds, rigorously extending prior work on transformer expressivity.
2. **Empirical Alignment**: Synthetic experiments validate theoretical predictions, with clear accuracy drops when CoT lengths are sublinear. Tests on real LLMs strengthen practical relevance, bridging theory and practice.
3. **Clarity**: The exposition is well-structured, with intuitive explanations of sensitivity and UHAT transformers, making complex theoretical arguments accessible.

### **Weaknesses**
1. **UHAT Assumption**: The reliance on *unique hard attention* limits direct applicability to real-world transformers, which use soft attention. While the authors justify UHAT for theoretical tractability, the practical implications for standard transformers remain limited.
2. **Scalability of Experiments**: Synthetic tasks use small inputs (e.g., 16-bit integers), raising questions about generalization to larger scales. While pragmatic for controlled validation, broader empirical tests (e.g., 64-bit MULTIPLICATION) could strengthen claims.

**Questions For Authors:**

Please refer to the weakness.

**Relation To Broader Scientific Literature:**

N/A

**Theoretical Claims:**

The theorems in paper make sense, and I believe they are correct. However, I didn't check the proof line by line.

---

> ### Author Rebuttal · Authors · 2025-03-31
>
> We thank the reviewer for the positive assessment of our experiments and theorems and pointing out the strengths of our paper. We now address the concerns raised in the review.
>
> 1. In Appendix A, we explain in detail why our results for hard attention are relevant for real-world transformers, especially concerning CoT. We briefly reiterate the arguments below.
>
>     First of all, many existing CoT constructions for soft attention in the literature (e.g., [1]) can be expressed in UHAT. Hence, even if lower bounds for soft attention don't directly follow from the lower bounds for UHAT, our impossibility results show that any sublinear solutions for considered tasks would likely to be hard to find.
>
>     Second, prior work [2] established a correspondence between the unlearnability of certain classes of tasks for soft attention and their impossibility for hard attention. Thus, our lower bounds for UHAT expressivity shed light on lower bounds for learnability in soft attention (and those are directly supported by our experiments).
>
> 2. In addition to our arguments in Appendix A, there is new substantial evidence proving a correspondence between hard and soft attention that has not yet been mentioned in our paper. Recent work, published after we had submitted our paper to ICML, rigorously establishes that some versions of UHAT are functionally equivalent to real-world Transformer setup (finite-precision causal soft attention) [3]. That result expands prior work that had already established that finite-precision soft attention is contained in $AC^0$ [4]. This directly establishes the relevance of our results for the real-world LLMs, as they *directly imply corresponding lower bounds for finite-precision soft-attention transformers*. We will make this implication explicit in the final version.
>
> 3. Regarding the scalability of our experiments, we agree that scaling our experiments would further illustrate our conclusions. Initially, we limited our multiplication experiments to 16 bits due to computational resource limitations. We will scale up our experiments, and include the results in the camera-ready version of the paper.
>
> 4. The reviewer has also mentioned that our template is wrong. We are not aware of any mismatch between our template and the official ICML template recommendations. We ask the reviewer to clarify what exactly is wrong with the template, and we can fix those issues in the final version of the paper.
>
> Finally, we kindly ask the reviewer to reconsider the score given to our paper. We would like to point out that the reviewer agrees that the paper is clear, experiments and theoretical results are thorough, and the claims are supported by convincing evidence. The only weaknesses mentioned are the UHAT assumption and the recommended scaling of the experiments, and we address those weaknesses above. Thus, we believe that the negative score assigned by the reviewer is unjustified given this list of strengths and weaknesses.
>
> ---
>
> [1] Abbe et al. "How far can transformers reason? the globality barrier and inductive scratchpad." NeurIPS 2024.
>
> [2] Hahn and Rofin. "Why are Sensitive Functions Hard for Transformers?" ACL 2024.
>
> [3] Jerad et al. "Unique Hard Attention: A Tale of Two Sides." arXiv preprint arXiv:2503.14615 (2025).
>
> [4] Li et al. "Chain of thought empowers transformers to solve inherently serial problems" ICLR 2024

---

### Decision · Program_Chairs · 2025-05-01

**Decision:**

Accept (poster)

**Comment:**

The paper presents lower bounds on the chain-of-thought (CoT) length required by unique hard-attention (UHAT) transformers to solve high-sensitivity algorithmic tasks (like median or reachability). Proof uses sensitivity theory and random restrictions, and bounds are tight up to logarithmic factors. This is a nice theoretical contribution to the literature on the expressive power of transformers. The model is mostly in line with what is used in practice, except for the use of unique hard attention instead of the commonly used soft attention. This is a restrictive assumption, but it is justified, given the difficulty of proving theoretical lower bounds on soft-max transformers. Overall, this is a nice theoretical contribution and can be accepted to ICML.